

# Using a multi-hypothesis framework to improve the understanding of flow dynamics during flash floods

Audrey Douinot[1], Hélène Roux[1], Pierre-André Garambois[2], and Denis Dartus[1]

[1]Institut de Mécanique des Fluides de Toulouse, IMFT, Université de Toulouse, CNRS - Toulouse, France
[2]Laboratoire des Sciences de l'ingénieur, de l'informatique et de l'imagerie (ICUBE) - INSA Strasbourg, Strasbourg, France

*Correspondence to:* Audrey Douinot (audreydouinot@gmail.com)

**Abstract.** A method of multiple working hypotheses was applied to a range of catchments in the Mediterranean area to analyse different types of possible flow dynamics in soils during flash flood events. The distributed, process-oriented model, MARINE, was used to test several representations of subsurface flows, including flows at depth in fractured bedrock, and flows through preferential pathways in macropores. Results show that the most realistic hypothesis for each catchment is consistent with in situ observations and measurements, when available. The study also highlights the potential of distributed modelling and spatial observations to deal with equifinality issues.

*Copyright statement.* The authors agree to the licence and copyright terms of Copernicus Publications as of 24 November 2017.

## 1  Introduction

### 1.1  Flash flood events: definition and description

Flash floods are defined as "sudden floods with high peak discharges, produced by severe thunderstorms that are generally of limited areal extent". (IAHS-UNESCO-WMO (1974); Garambois (2012); Braud et al. (2014)). They are often linked to localised and major forcings (greater than 100mm, Gaume et al. (2009)) at the heads of steep-sided, meso-scale catchments (with surface areas of 10-250 km$^2$).

In Europe, particularly intense flash floods are observed predominantly on the north west of the Mediterranean Arc, at the level of the mountain foothills. The regions affected are highly specific and marked by the influence of the Mediterranean climate system and mountainous topography. The steep topography and small size of the areas involved explain the rapid responsiveness of the catchments. The orographic effects on atmospheric circulation result in a higher accumulation of precipitation and localised convection cells (Marchi et al., 2010; Garambois et al., 2014). Flash floods are, thus, the result of particular hydrological (or physiographic) and meteorological conditions (Collier, 2007).

The large specific discharges, and intensities of precipitation, lead to the flash floods that occur being classified as extreme. However, this does not necessarily mean their occurrence is exceptional: on average, there were no fewer than five flash floods a year on the Mediterranean Arc between 1958 and 1994 (Jacq, 1994). The EM-DAT (International Disaster Database, CRED



2013), which records natural disasters affecting populations worldwide, also reports 33 thunderstorm episodes in Europe over the last ten years. Moreover, the first observations of global warming on the Mediterranean Arc signal an increase in the frequency and/or severity of events (Llasat et al., 2014; Colmet Daage et al., 2016).

### 1.2 Flash flood events: an issue for forecasters

5 Flash floods constitute a significant hazard and, therefore, a considerable risk for populations. In general, floods, and the flooding they can cause, represent the world's principal natural hazard (UNISDR 2009). Every year, 280 floods or storms are recorded as being disasters worldwide; whereas, statistically, over the same period, 31 earthquakes and 6 volcanic eruptions will have affected a population somewhere (Llasat et al., 2014). One of the main explanatory factors is the vulnerability of the areas prone to flooding, which are undergoing increasing urbanisation.

10 Flash floods are particularly dangerous due to their characteristics: (i) the suddenness of events makes it difficult to warn populations in time, and can lead to panic, thus increasing risk, when a population is unprepared, ii) the magnitude of floods implies significant amounts of kinetic energy, which can transform transitory rivers into torrents, resulting in the transport of debris ranging from fine sediments to tree trunks, as well as the scouring of river beds and the erosion of banks.

A major area of interest for flash floods is, therefore, better risk assessment, to enable them to be forecasted and the relevant 15 populations to be pre-warned. However, this is not an easy task, because most of the small catchments concerned do not have gauges installed, and they therefore, cannot be connected to an automatic monitoring system. Moreover, weather forecasts remain uncertain, with regard to the intensity of precipitation and, above all, of the location of rain cells. Their use is therefore problematic, especially at the scale of these small catchments.

Greater knowledge and understanding is required to better identify the determining factors that result in flash floods. In 20 particular, in order to implement a regional forecasting methodology, the properties of the catchments, and the climatic forcing and linkages between them which lead to flash flood events need to be characterised.

### 1.3 Flash flood events: understanding flow processes

Due to the challenges involved in forecasting flash floods, especially against a background of climate change which is tending to amplify the phenomenon (Llasat et al., 2014; Colmet Daage et al., 2016), there has been considerable research done on 25 the subject over the last ten years. Examples include the HYDRATE project (2006-2010, Gaume and Borga (2013)), which enabled the setting up of a comprehensive European database of flash flood flash events, as well as the development of a reference methodology for the observation of post-flood events; the EXTRAFLO project (2009-2013, Lang et al. (2014)) to estimate extreme precipitation and floods for French catchments; the HYMEX project (2010-2020, Drobinski et al. (2014)) focusing on the meteorological cycle at the Mediterranean scale, and, in particular, on the conditions that allow extreme events 30 to develop; or the FLOODSCALE project (2012-2016, Braud et al. (2014)), based on a multi-scale experimental approach to improve observation of the hydrological processes that lead to flash floods.

This latest research demonstrates, in particular, the importance of cumulative rainfall (Arnaud et al., 1999; Sangati et al., 2009), previous soil moisture state (Cassardo et al., 2002; Marchandise and Viel, 2009; Hegedüs et al., 2013; Mateo Lázaro et al.,



2014; Raynaud et al., 2015) and the storage capacity of the area affected by the precipitation (Viglione et al., 2010; Zoccatelli et al., 2010; Lobligeois, 2014; Garambois et al., 2015a; Douinot et al., 2016). The combined influence of the spatial distribution of precipitation and event-related storage capacities, reported in the study of a number of particular events (Anquetin et al., 2010; Le Lay and Saulnier, 2007; Laganier et al., 2014; Garambois et al., 2014), suggests a hydrological reaction, in some areas of

the catchments, that arises from localised soil saturation. This statement surmises that there is little direct Hortonian flow, but rather the production of runoff through excess soil saturation, or lateral fluxes in the soil resulting from the activation of preferential pathways.

The geochemical monitoring of eight intense precipitation events, over a 3.9 km$^2$ catchment area, during the FLOODSCALE project (Braud et al., 2014), revealed a "flushing" phenomenon. In at least the first 40 cm of the soil layer, the water present

at the start was replaced by so-called "new" rainwater (Braud et al., 2016; Bouvier et al., 2017). The proportion of new water at the peak of the flood varied between $50\%$ and $80\%$ depending on the intensity of precipitation and the moisture level at the start of the event. Conversely, over the entire period of the event, it appears that new water accounts for only between $20\%$ and $30\%$ of the total volume of water discharged, which underlines the dominance of intra-soil dynamics.

Being able to define the storage capacity of the soil column is crucial in explaining the varied responses of the catchments.

Geological properties, which are crucial physiographic characteristics for determining the total storage capacity of catchments (Sayama et al., 2011; Pfister et al., 2017a), also appear to be markers of the storage capacities available over the time scales involved in flash floods (which are of the order of a day). From simple flow balances of flash flood events (Payrastre et al., 2012), studies of the diverse hydrological responses of several catchments over the same precipitation episode (Douinot, 2016a), or the application of regional hydrological models (Garambois et al., 2015b), the literature tends to demonstrate the low storage

capacity of non-karst sedimentary and marl-type catchments, and, conversely, the potential for storing large volumes of water in the altered rocks of granitic or schist formations. Flow dynamics during flash floods thus appear to depend on the hydrogeological functioning of the catchments which again emphasises the importance of the saturation dynamics of the "soil + altered substratum" combination.

### 1.4 The potential of a multi-model study for understanding hydrological behaviour

The knowledge gained about the development of the flow processes (for example, the tracing of events carried out during the FLOODSCALE project, Braud et al. (2014)), relates to studies on a number of specific sites where flash floods could be observed while they were taking place. However, being able to generalise the knowledge gained is limited by the specific nature of each study (McDonnell et al., 2007) and by the gap between the spatial scale of forecasts (meso-scale), compared with that of the in-situ observations (<10 km$^2$) (Sivapalan, 2003). Such hydrological modelling work can be considered as a

means of extrapolating knowledge to an extended geographical area, possibly covering catchments with differing physiographic properties.

Moreover, hydrological models viewed as "tentative hypotheses about catchment dynamics" are interesting tools for testing hypotheses about hydrological functioning using a systematic methodology. A considerable amount of recently published work has involved comparative studies, using numerical models to develop or validate the hypotheses about the type of hydrological





functing that is most likely to reproduce hydrological responses accurately (Buytaert and Beven, 2011; Clark et al., 2011; Fenicia et al., 2014; Coxon et al., 2014; Ley et al., 2016; Fenicia et al., 2016). For example, Fenicia et al. (2014) show that the performance of different models tested on the Attert Basin in Luxembourg corroborate the various hydrological processes known to occur in this catchment; non-linear models are better for modelling the hydrological dynamics of drainage sub-catchment basins on impermeable bedrock layers and those exhibiting threshold behaviour; conversely linear models with parallel storage elements led to better reproduction of the hydrological signature of the catchments with smoother responses.

The principle of "the method of multiple working hypotheses" is to compare the results from models governed by different assumptions about hydrological processes. Comparisons are even more meaningful if the structure of the models compared differs solely in terms of the hypotheses tested, in the form of modules. Doing this avoids the limitations on interpretation that are often encountered in comparative studies of models (Perrin et al., 2013), where numerical choices can influence results independently of the underlying assumptions. The comparative study makes it possible to conclude either a known hydrological functioning, which is distinguished by the better performance of the inherent model, or indeterminacy in the case of an equivalent fit of the models. The equifinality of the models remains instructive because it makes it possible to detect the underlying uncertainties behind the hypothesis of the models, which then helps determine avenues for further research.

The multiple working hypotheses framework is usually applied using a flexible conceptual and lumped model framework, such as the FUSE (Clark et al., 2008) or SUPERFLEX (Fenicia et al., 2011). It is related to continuous hydrological responses in order to assess hydrological hypotheses through the overall hydrological signature of the catchments. **In this work, we extend the method of multiple working hypotheses to distributed, mecanistic and event-based hydrological models. The objective is to test a number of proposed hydrological functioning that occur during flash flood events on a set of contrasting catchments in the French Mediterranean area.**

## 1.5 Current issues, objectives and plan

Other than the observations discussed above, which were made on a specific small site ($<10$ km$^2$), there is little information on the formation of flows in the soil and/or geological layers. While the proportion of flows passing through the soil appears to be significant, questions arise about how they form:

– Are they subsurface flows that take place in a restricted area of the root layer, as a result of preferential path activation? Or, are they lateral flows taking place at greater depth comparable to those seen in some aquifer?

– Does the geological bedrock or an altered substratum play a role limited to that of mere storage reservoir, or is it actively involved in flood flows formation?

– Can the hydrological processes be discerned from the nature of the geological bedrock?

The aim of this article is to attempt to answer these questions using a multi-model approach that tests different types of hydrological dynamics. The study was based on MARINE, a physically based, distributed hydrological model (Roux et al., 2011; Garambois et al., 2015a), which was developed specifically to model flash floods in the catchments of the French Mediterranean



Arc. Several new representations for the soil column and underground flows are proposed (Douinot, 2016b) and included in the MARINE model, in the form of modules that can be used to test different hydrological functions. Those different hydrological dynamics are applied to a set of catchments with physiographic properties representative of the whole of the French Mediterranean Arc. The performance of each model are then examined and subjected to a comparative study.

The structure of the publication is as follows: Section 2 describes the catchments and different datasets used in the study. Section 3 describes the MARINE model and the hypotheses about flow dynamics that were tested. Section 4 describes the evaluation methodology used to characterise the performance of each model. Section 5 presents the key results of the study, in the form of a comparative description of the simulations that resulted from the different modelling choices made. Lastly, the final section sets out conclusions and discusses the works contribution to our understanding of the hydrological functioning of

catchments during flash floods and the effectiveness of the methodology adopted.

## 2   Catchments and data used in the study

### 2.1   Description of the catchments used in the study

We studied the behaviour of four catchments and eight nested catchments in the French Mediterranean Arc (Figure 1). The catchments (in the order they are numbered in Figure 1) were those of the Ardèche, Gardon, Hérault and Salz rivers; these

were selected for the following reasons: (i) they are representative of the physiographic variability found in areas where flash floods occur; (ii) numerous studies of flash floods have already been carried out on the Gardon and Ardèche (Ruin et al., 2008; Anquetin et al., 2010; Delrieu et al., 2005; Maréchal et al., 2009; Braud et al., 2014), for example. Knowledge of the hydrological functioning of these catchments could guide the interpretation of the modelling results (Fenicia et al., 2014); and (iii) a considerable number of observations of flash flood events are available for these catchments.

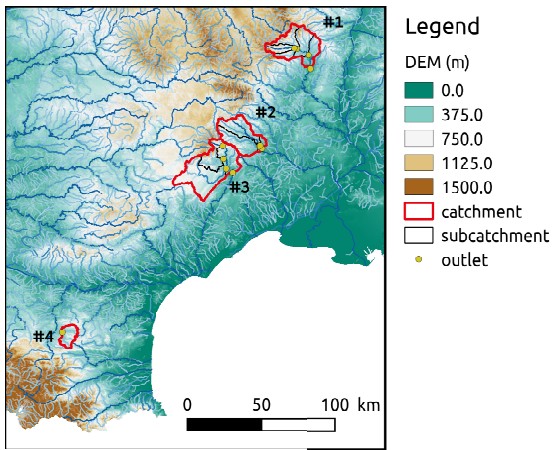

**Figure 1.** Locations of the catchments studied, with a topographic visualisation at $25\ m$ resolution (Source: IGN, MNT BDALTI)





**The Ardèche catchment** at Vogüe has a surface area of 622 km$^2$. We also studied the behaviour of sub-catchments at Meyras (99 km$^2$), Pont-la-Beaume (292 km$^2$) and Ucel (477 km$^2$). The Ardèche catchment upstream of Ucel sits essentially on a granite bedrock with some sandstone on its edges. Downstream, the geology changes to a predominantly schist and limestone formations (Figure 2). In this area, studies from experimental sites show that flows are mainly due to surface runoff from cultivated soils (Braud and Vandervaere, 2015). The mostly sand-loam soils, covering the entire catchment area, are relatively deep (47 cm) and become shallower as the elevation increases.

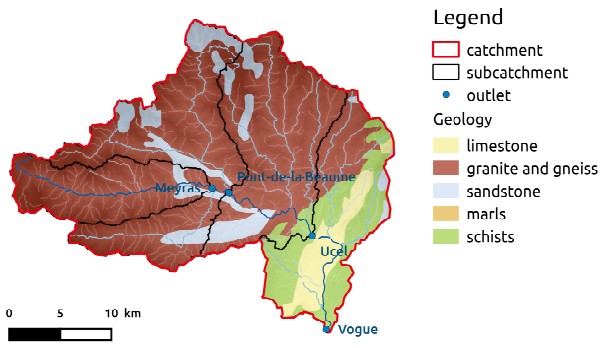

**Figure 2.** The geology of the Ardèche (sources : BD Million-Géol, BRGM)

**The Gardon catchment** at Anduze has a surface area of 543 km$^2$. We also studied the behaviour of the sub-catchments at Corbès (220 km$^2$) and Mialet-Roucan (240 km$^2$), which are two separate sub-catchments. The Gardon catchment is marked by clear upstream/downstream differences (Figure 3). The upstream consists of schistose bedrock, and mainly silty soil of shallow depth. Downstream, the bedrock is impermeable marl-type and granite formation, with the latter assumed to be altered. The soil there can be more than a metre deep. Observations of the hydrological functioning of a number of catchments, carried out over surface areas of the order of one km$^2$, (Ayral et al., 2005; Maréchal et al., 2009; Martin et al., 2005; Maréchal et al., 2013) show that, for the schistose part, flows seem to form rapidly mainly in the subsurface, while on the granitic part of the catchment, flow formation appears to be controlled by the extension of the saturated zone related to the river.

**The Hérault catchment** at Laroque has a surface area of 912 km$^2$. The behaviour of the sub-catchments of Saint-Laurent-le-Minier (499 km$^2$), La Terrisse (155 km$^2$) and Valleraugue (46 km$^2$) were also studied. The Hérault catchment has highly contrasting physiographic properties, which are highlighted when it is split into sub-catchments. The sub-catchments at Valleraugue and La Terrisse are on the Cévennes Massif. They sit mainly on schists, but also on granite and gneiss. The catchments are very steep, particularly upstream of Valleraugue, and the soil is mostly silty. Conversely, the sub-catchment upstream of Saint-Laurent-le-Minier is predominantly a limestone plateau, and the slopes are less steep and covered with a silt-loam soil with less capacity for infiltration. The presence of a large karst formation, revealed in particular by a less developed surface hydrographic network (Figure 3), should be noted on this sub-catchment. As a result of the physiographic diversity, there are considerable differences between the mean hydrological responses of the sub-catchments (Table 1).




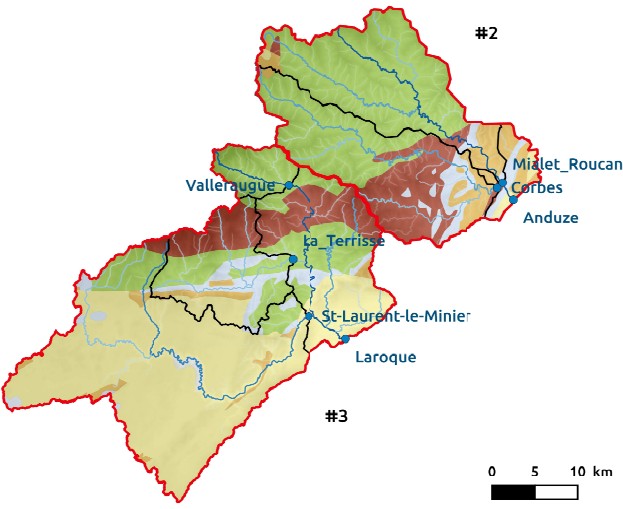

**Figure 3.** The geology of the Gardon and Hérault catchments (sources : BD Million-Géol, BRGM)

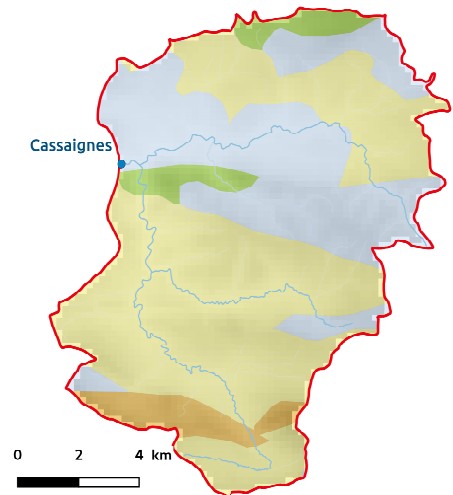

**Figure 4.** The geology of the Salz catchment (sources : BD Million-Géol, BRGM)

**The Salz catchment** at Cassaigne has a surface area of 144 km$^2$. It is representative of the catchments found in the Corbières (foothills of the Pyrenees), an area frequently affected by flash floods. It is characterised by sedimentary bedrock comprising sandstone and limestone (figure 4). The slopes of this catchment are less steep than the other catchments studied. Conversely, soils are relatively deep, and the low mean inter-annual discharge is indicative of a low base flow.

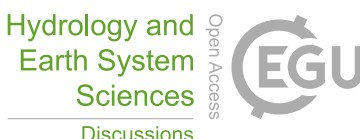


Table 1 summarises the main geological, soil and topographical characteristics of the catchments studied.

**Table 1.** Physiographic properties and hydrological statistics of the 12 catchments *ID*: coding name of the catchments used at figure 1; area [km²]; mean slope [-]; mean soil depth [m], percentage of bedrock geology [%] including sandstone (Sa), limestone (Li), granite and gneiss (GG), marls (Ma) and schists (Sc) subcategories; main soil texture (Tx); mean annual precipitation ($P[mm]$) ; mean inter-annual discharge ($Q[m^3.km^{-2}.s^{-1}]$); 2 year return period daily discharge ($Q_{D2}[m^3.km^{-2}.s^{-1}]$); 10 year return period hourly discharge ($Q_{H10}[m^3.km^{-2}.s^{-1}]$). Hydrometric statistics are calculated from HydroFrance databank, (Ministère de l'Ecologie, 2015) and the pluviometric ones using rainfall data from the raingauge network of the French flood forecasting services.

| ID | River | Outlet | Area [km²] | Slope [-] | Soil depth [m] | Sa [%] | Li [%] | GG [%] | Ma [%] | Sc [%] | Tx | P[mm] | Q | $Q_{D2}$ | $Q_{H10}$ |
|---|---|---|---|---|---|---|---|---|---|---|---|---|---|---|---|
| #1 | L'Ardèche | Vogue | 622 | 0.17 | 0.47 | 10.5 | 5.7 | **71.9** | 0.0 | 11.9 | Ls | 1587 | 0.041 | 0.62 | 2.25 |
| | | Ucel | 477 | 0.20 | 0.45 | 13.7 | 0.0 | **84.5** | 0.0 | 1.8 | Ls | 1577 | 0.046 | 0.79 | 2.30 |
| | | Pont de la Beaume | 292 | 0.22 | 0.39 | 14.0 | 0.0 | **86.0** | 0.0 | 0.0 | Ls | 1690 | 0.056 | 0.75 | 2.53 |
| | | Meyras | 99 | 0.24 | 0.32 | 5.4 | 0.0 | **94.6** | 0.0 | 0.0 | Ls | 1720 | 0.036 | 0.72 | 2.92 |
| #2 | Le Gardon | Anduze | 543 | 0.16 | 0.25 | 7.2 | 1.5 | 18.0 | 12.1 | **61.2** | L | 1370 | 0.026 | 0.48 | 1.82 |
| | | Corbès | 220 | 0.16 | 0.27 | 9.3 | 0.0 | 34.2 | 9.0 | **47.5** | L | 1460 | 0.022 | 0.57 | 2.28 |
| | | Mialet Roucan | 240 | 0.17 | 0.22 | 2.0 | 0.6 | 2.9 | 9.4 | **85.1** | L | 1407 | 0.023 | 0.62 | 2.54 |
| #3 | L'Hérault | Laroque | 912 | 0.14 | 0.26 | 6.7 | **54.5** | 11.7 | 3.2 | 24.0 | Lsi | 1160 | 0.019 | 0.39 | 1.21 |
| | La Vis | St Laurent le Minier | 499 | 0.10 | 0.26 | 4.0 | **83.0** | 1.0 | 3.2 | 8.8 | Lsi | 930 | 0.018 | 0.42 | 1.10 |
| | L'Arre | La Terrisse | 155 | 0.19 | 0.25 | 19.5 | 12.3 | 27.2 | 6.2 | **34.8** | L | 1130 | 0.027 | 0.61 | 2.0 |
| | L'Hérault | Valleraugue | 46 | 0.27 | 0.25 | 0.0 | 0.0 | 0.0 | 0.0 | **100.0** | L | 1920 | 0.049 | 1.13 | 4.0 |
| #4 | La Salz | Cassaigne | 144 | 0.13 | 0.37 | 33.5 | **56.5** | 0.0 | 5.1 | 4.9 | Lsi | 700 | 0.008 | 0.20 | 1.31 |



## 2.2 Forcing inputs and hydrometric data

The hydrometric data were derived from the network of operational measurements (HydroFrance databank, Ministère de l'Ecologie (2015)). Eight to twenty years of hourly discharge observations were available, according to the dates when the hydrometric stations were installed.

Flood events with peak discharges that had exceeded the 2-year return period daily discharge ($Q_{D2}$, in Table 1, corresponds to the alert threshold for flood forecasting centres in France) were selected as events to be included in the study. Thus, only one criterion for hydrological response was considered. This led to a selection of precipitation events of varying origins (for instance: rainfall induced by mountains, stagnant convective cells; and rainfall occurring in different seasons - mainly in autumn and early spring). Such a selection risked complicating the study because flow processes can vary from one season to another.

Nevertheless, it allowed us to test the ability of the model to deal with different (non linear) flow physics regimes. The aim of this selection was to be able to analyse, more broadly, overall catchment behaviour during intense events.

    Precipitation measurements were taken from Météo France's ARAMIS radar network (Tabary, 2007), which provides precipitation measurements, at a resolution of $1\ km \times 1\ km$, every five minutes. These measurements were calibrated by forecasters at the French Flood Forecasting Service by monitoring a network of rain gauges using RHEA's CALAMAR software. Depend-

ing on the availability of the results of rainfall and hydrometric measurements, 7 to 14 intense events were selected for each catchment (Table 2). Table 2 lists the mean properties of the events selected, such as cumulative precipitation during the event or peak flow.

    Some differences in meteorological forcing and the hydrological responses of catchments can be noted. The Ardèche (♯1) is subject to more significant events in terms of cumulative precipitation, with a notable orographic gradient. In contrast,

cumulative precipitations in the Salz catchment (♯4) are the lowest. The highest precipitation intensities have been recorded in the Gardon catchment (♯2). The events selected on this catchment cover a wide range of peak flows despite relatively uniform cumulative precipitation. The Hérault catchments (♯3) at Laroque and Saint Laurent le Minier had more uniform hydrological responses for meteorological forcing similar to that of the Gardon catchment in terms of precipitation, but these were lower in intensity.

As the MARINE model is event-based, it must be initialised to take into account the previous moisture state of the catchment, which is linked to the history of the hydrological cycle. This was done using spatial model outputs from Météo-France's SIM operational chain (Habets et al., 2008). Based on the work of Marchandise and Viel (2009), the spatial daily root-zone humidity outputs (resolution = $8\ km \times 8\ km$) simulated by the SIM conceptual model were used for the systematic initialisation of MARINE.



**Table 2.** Properties of the flash flood events *ID*: coding name of the concerned catchments (figure 1 : ♯1 for the Ardèche; ♯2 for the Gardon; ♯3 for the Hérault and ♯4 for the Salz); $N_{evt}$: number of observed flash flood events; P [mm] mean precipitation (± standard); $I_{max}[mm.h^{-1}]$ : mean of the maximal intensity rainfall per event; $Q_{peak}$: specific flood peak $[m^3.km^{-2}.s^{-1}]$; Hum : initial soil moil moisture according to SIM output (Habets et al., 2008) [%]

| ID | Outlet | $N_{evt}$ | P [mm] | $I_{max}[mm.h^{-1}]$ | $Q_{peak}[m^3.km^{-2}.s^{-1}]$ | Hum [%] |
|---|---|---|---|---|---|---|
| ♯1a | Vogue | 10 | 192 (±93) | 17.3 (±6.2) | 1.33 (±0.57) | 58 (±6) |
| ♯1b | Ucel | 10 | 208 (±105) | 19.1 (±7.1) | 1.41 (±0.70) | 56 (±5) |
| ♯1c | Pont de la Beaume | 10 | 222 (±122) | 20.5 (±6.2) | 1.79 (±0.82) | 56 (±5) |
| ♯1c | Meyras | 10 | 235 (±141) | 25.6 (±10.6) | 2.15 (±1.15) | 56 (±4) |
| ♯2a | Anduze | 13 | 182 (±69) | 26.9 (±12.6) | 2.10 (±1.67) | 53 (±7) |
| ♯2b | Corbès | 14 | 196 (±73) | 31.4 (±11.6) | 1.90 (±0.93) | 55 (±7) |
| ♯2c | Mialet Roucan | 14 | 177 (±72) | 30.9 (±13.2) | 1.85 (±0.85) | 51 (±7) |
| ♯3a | Laroque | 7 | 188 (±95) | 16.0 (±8.1) | 0.82 (±0.43) | 59 (±8) |
| ♯3b | St Laurent le Minier | 7 | 153 (±95) | 18.4 (±8.9) | 1.14 (±0.31) | 56 (±9) |
| ♯3c | La Terrisse | 7 | 193 (±103) | 22.1 (±12.1) | 1.63 (±0.87) | 52 (±8) |
| ♯3d | Valleraugue | 7 | 156 (±110) | 16.4 (±8.7) | 2.14 (±1.33) | 48 (±6) |
| ♯4 | Cassaigne | 8 | 136 (±47) | 17.8 (±6.2) | 1.48 (±0.64) | 57 (±7) |

## 3 The multi-hypothesis hydrological modelling framework

### 3.1 The MARINE framework

The MARINE model is a distributed mecanistic hydrological model specially developed for flash flood simulations. It models the main physical processes in flash floods: infiltration, overland flow, lateral flows in soil and channel routing. Conversely, it

does not incorporate low-rate flow processes such as evapotranspiration or base flow.

MARINE is structured into three main modules that are run for each catchment grid cell (see Figure 5). The first module allows the separation of surface runoff and infiltration using the Green-Ampt model. The second module represents subsurface downhill flow. It was initially based on the generalised Darcy Law used in the TOPMODEL hydrological model (Roux et al., 2011), but was developed in greater detail as part of this study. Lastly, the third module represents overland and channel

flows. Rainfall excess is transferred to the catchment outlet using the Saint-Venant equations simplified with kinematic wave assumptions. The model distinguishes grid cells with a drainage network (where channel flow is calculated on a triangular channel section (Maubourguet et al., 2007)) from grid cells on hillslopes (where overland flow is calculated for the entire surface area of the cell).

The MARINE model works with distributed input data such as: i) a digital elevation model (DEM) of the catchment to

15 shape the flow pathway and distinguish hillslope cells from drainage network cells, according to a drained area threshold; ii) soil survey data to initialize the hydraulic and storage properties of the soil, which are used as parameters in the infiltration and



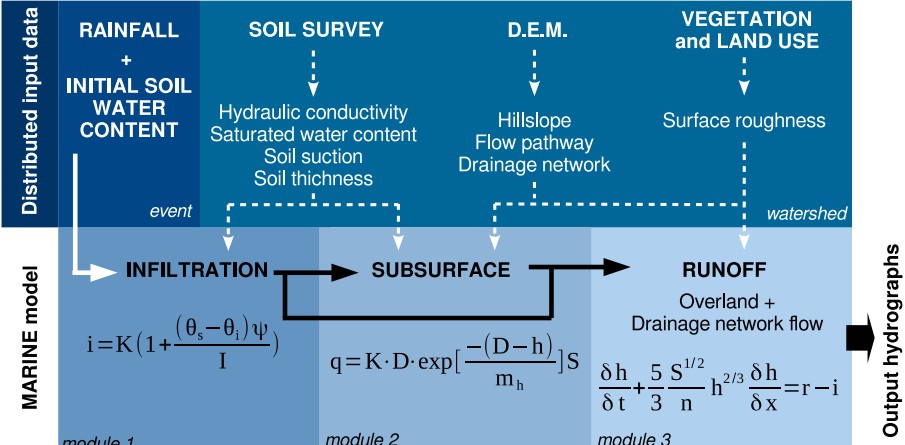

**Figure 5.** The MARINE model structure, parameters and variables. The Green and Ampt infiltration equation contains the following parameters: infiltration rate i (m.s$^{-1}$), cumulative infiltration I (mm), saturated hydraulic conductivity k (m.s$^{-1}$), soil suction at the wetting front $\Psi$ (m), and, current and initial water contents, $\theta_s$ and $\theta_i$ (m$^3$.m$^{-3}$), respectively. Subsurface flow contains the following parameters: soil thickness (m), lateral saturated hydraulic conductivity K (m.s$^{-1}$), local water depth h (m), transmissivity decay with depth $m_h$ (m), and bed slope S (m.m$^{-1}$). The kinematic wave contains the following parameters: surface water depth h (m), time t (s), space variable x (m), rainfall rate r (m.s$^{-1}$), infiltration rate i (m.s$^{-1}$), bed slope S (m.m$^{-1}$), Manning roughness coefficient n (m$^{-1/3}$.s). The Module 2 described in this figure corresponds to the standard definition applied to the MARINE model. It corresponds, in fact, to the scope of model modifications proposed in this study, which are described in the next section (section 3.2.)

lateral flow models; iii) vegetation and land-use data to configure the surface roughness parameters used in the overland flow model.

The MARINE model requires parameters to be calibrated in order to be able to reproduce hydrological behaviours accurately. Based on sensitivity analyses of the (Garambois et al., 2013) model, five parameters are calibrated: soil depth - $C_z$, the
5  saturation hydraulic conductivity used in lateral flow modelling - $C_{kss}$, hydraulic conductivity at saturation, used in infiltration modelling - $C_k$, and friction coefficients for low and high-water channels - $n_r$ and $n_p$, respectively, with $n_r$ and $n_p$ uniform throughout the drainage network. $C_{kss}$, $C_k$ and $C_z$ are the multiplier coefficients for spatialised, saturated hydraulic conductivities and soil depths. In this study, it was specifically Module 2 that was subject to modifications in order to determine the possible ways that a number of proposals for intra-soil hydrological functioning could be modelled. To do this, modifications
10  were made to the parameters $C_z$ and $C_{kss}$.

## 3.2 Modelling lateral flows in the soil: the development of a multi-hypothesis framework

The role of altered rocks has been demonstrated in the previous work of (Payrastre et al., 2012; Vannier et al., 2013; Garambois et al., 2015b). The integration of this hydrologically active zone into MARINE was done by the calibration of $C_z$: soil-depth data from the BDsol databases (Robbez-Masson et al., 2002) are artificially increased to take account of the substratum.



Here, the aim was, specifically, to integrate hydrological activity at depth, especially given that it seems to differ according to the geological properties of the bedrock (Fenicia et al., 2014; Pfister et al., 2017a). We proposed a number of modifications to Module 2 covering three hypotheses about hydrological functioning:

- Deep Water Flow model (DWF): we assumed deep infiltration and the formation of an aquifer flow in highly altered rocks. In hydrological terms the pedology-geology boundary was transparent. The soil column could be modelled as a single entity of depth $D_{tot}$ $(m)$, which is at least equal to the soil depth $D_{BDsol}$ $(m)$ (see Figure 6). Given the lack of knowledge and available observations, a uniform calibration was applied to the depth of altered rocks - $D_{WB}$ $(m)$ - a level that is rapidly accessible on the scale of a rain event. Groundwater flow was described using the generalised Darcy Law ($q_{dw}$, Equation 1). The exponential growth of the hydraulic conductivity at saturation, as the water table ($h_{dw}$) rises (the TOPMODEL approach), assumed an altered-rock structure where hydraulic conductivity at saturation decreases with depth.

- Subsurface Flow model (SSF): We assumed that the formation of subsurface lateral flows was due to the activation of preferential paths, like the in-situ observations of Katsura et al. (2014) and Katsuyama et al. (2005). The altered soil-rock interface acts as a hydrological barrier. The rapid saturation of shallow soils results in the development of rapid flows due to the steep slopes of the catchments and the existence of rapid water flows circulating through the macropores as the soil becomes saturated. The soil column was thus represented by a two-layer model (see Figure 7): an upper layer of depth equal to the soil depth $D_{BDsol}$ $(m)$ and a lower layer of uniform depth $D_{WB}$ $(m)$. The lateral flows in the upper layer were described by the generalised Darcy Law. However, variations in hydraulic conductivity were expressed as a function of the mean water content of the layer ($\theta_{soil}$) and not of the height of water ($h_{soil}$) that would form a perched water table (Equation 2). Expressing the variability in hydraulic conductivity as a function of the saturation rate indeed appears to be a more appropriate choice for representing the activation of preferential paths in the soil by the increase in the degree to which the soil is filled. The decay factor of the hydraulic conductivity as a function of the saturation rate - $m_\theta$ - was set according to the linearized empirical relations, developed by Van Genuchten (1980), between hydraulic conductivity and soil water content for the different classes of soil textures. Flows in the lower soil layer ($q_{dw}$, Equation 3), in the form of a deep aquifer, were limited by setting the hydraulic conductivity of the substratum as being equivalent to that of the soil divided by 50 (this choice being guided by the orders of magnitude generally observed in the literature (Le Bourgeois et al., 2016; Katsura et al., 2014)). The altered rocks were thus assumed to play, mainly, a storage role. Infiltration occurring between the two layers was initially restricted by the Richards equations which were incorporated using the set hydraulic properties of the substratum (Equation 4). When the upper layer is saturated, filling by a piston effect is allowed. The depth of the soil layer, $D_{BDsol}$, was set according to the soil data, while the depth of the substratum - $D_{WB}$ - was calibrated in the same way as in the DWF model.

- The Subsurface and Deep Water Flow model (SSF-DWF): It was assumed that the presence of subsurface flow was due to both local saturation of the top of the soil column, but also the development of a flow at depth, as a result of significant volumes of water introduced by infiltration and a very altered substratum whose apparent hydraulic conductivity was





already relatively high. This hypothesis of the process led to a modelling approach analogous to the SSF model (Figure 7), where the hydraulic conductivity at substrate saturation - $K_{dw}$ - was no longer simply imposed, but, instead, calibrated using an additional coefficient, $C_{kdw}$.

The soil water content prior to simulation was, similarly, initialised for each model, in order to ensure, for a fixed depth of altered rock, that the same volume of water was allocated for all models. The SIM humidity indices (Section 2.2) were used to set an overall water content for all groundwater flow models for a given flood, with the two compartments of the SSF and SSF-DWF models then having an equal water content at initialisation.

## 4 Methodology for calibrating and evaluating the models

### 4.1 Calibration method

The three hydrological models studied - DWF, SSF and SSF-DWF - were calibrated for each catchment by weighting 5,000 randomly drawn samples from the parameter space for each model (the Monte Carlo Method). The weighting was done using the DEC (Discharge Envelope Catching) score (Equation 5), discussed by Douinot et al. (2017), in order to integrate the a priori uncertainties of modelling $\left( (\sigma_{mod,i}),\ i = 1...n \right)$ (cf. eq. 6) and those related to the flow measurements $\left( (\sigma_{\hat{y}_i}),\ i = 1...n \right)$ (cf. eq. 7). The choice of DEC is justified by the desire to adapt the evaluation criterion to the modelling objectives (for example, by focusing calibration on reproduction of the rise and peaks of floods in order to be able to forecast flash floods) while always being aware of the uncertainties in the reference flow measurements.

Given the lack of information, these uncertainties $\left( (\hat{y}_i),\ i = 1...n \right)$ were set at 20 % of the measured discharge, which is in line with the literature on discharge measurements from operational stations (Le Coz et al., 2014), and increased linearly with the 10-year hourly discharge, beyond which, as a general rule, the observed flow is no longer measured, but derived by extrapolation from a discharge curve, making it less accurate (Equation 7).

The modelling uncertainties $\left( (\sigma_{mod,i}),\ i = 1...n \right)$ were set at a minimum value - as a function of the basic catchment module, thus ensuring that the evaluation of the hydrographs would not be unduly affected by the reproduction of relatively low flows which were strongly dependent on initialisation using previous moisture data that were not the subject of this study. In addition, it was assumed that a modelling uncertainty of 10 % around the confidence interval of observed flows was acceptable (Equation 6).

$$
\begin{aligned}
DEC &= \frac{1}{n} \sum_{i=1}^{n} \frac{d_i}{\sigma_{mod,i}} & (5) \\
\sigma_{mod,i} &= 0.5 * Q + 0.025 * \hat{y}_i & (6) \\
\sigma_{\hat{y}_i} &= 0.05 * \hat{y}_i * \left( 1 + \frac{\hat{y}_i}{Q_{H10}} \right) & (7)
\end{aligned}
$$

with $\hat{y}_i$ and $\sigma_{\hat{y}_i}$ the observed discharge and the uncertainty of measurement at time $i$; $d_i$ the discharge distance between the model prediction at time $i$ ($y_i$) and the confidence interval of the discharge measurement (that is to say the distance of $y_i$ to

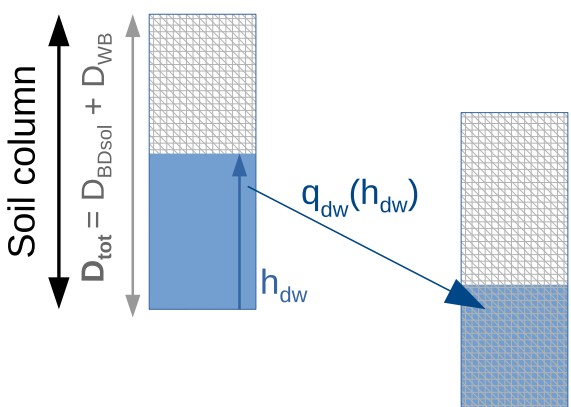

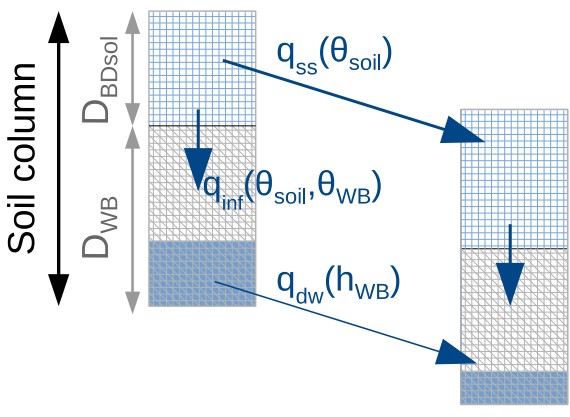

**Figure 6.** DWF model: flow generation by infiltration at depth and support of a deep aquifer ($q_{dw}$).

**Figure 7.** SSF and SSF-DWF models: flow generation by the saturation of the upper part of soil column and activation of preferential paths ($q_{ss}$). Supported flow at depth - $q_{dw}$ - is greater or lesser according to the storage capacity of the substratum.

$$q_{ss} = K_{ss} \cdot D_{BDsol}\, exp\left(\frac{\theta_{soil} - 1}{m_\theta}\right) \cdot S \qquad (2)$$

$$q_{dw} = K_{dw} \cdot D_{WB}\, exp\left(\frac{h_{WB} - D_{WB}}{m_h}\right) \cdot S \qquad (3)$$

$$q_{inf} = -K_{dw}\frac{\delta H(\theta_{soil}, \theta_{WB})}{\delta z} \qquad (4)$$

$h_{soil}, h_{WB}[m]$: soil water depth in the upper and lower layer respectively

$\theta_{soil}, \theta_{WB}[-]$: soil water content of the upper and lower layer respectively

$m_\theta[-]$: decay parameter of the hydraulic conductivity with soil water content $\theta_{soil}$

$K_{ss} = C_{kss} \cdot K_{BDsol}$

$K_{dw} = 0.02 \cdot K_{ss}$ in SSF model and

$K_{dw} = C_{kdw} \cdot K_{BDsol}$ in SSF-DWF model

$$q_{dw} = K_{dw} \cdot D_{tot}\, exp\left(\frac{h_{dw} - D_{tot}}{m_h}\right) \cdot S \qquad (1)$$

$h_{dw}[m]$ : water depth of the unique water table

$m_h[m]$ : decay parameter of the hydraulic conductivity with hydraulic head

$S[-]$ : bed slope

$K_{dw} = C_{kdw} \cdot K_{BDsol}$

$D_{tot} = D_{BDsol} + D_{WB}$

$[\hat{y}_i - \sigma_{\hat{y}_i}, \hat{y}_i - \sigma_{\hat{y}_i}]$); $\sigma_{mod,i}$ the simulated uncertainty at time $i$; $Q$ and $Q_{H10}$ respectively the mean inter-annual discharge and the 10-year hourly discharge of the related catchment.





## 4.2 Metrics and key points in model evaluation and comparison

The objective was to evaluate the fit of the models in terms of reproducing the different phases of the hydrographs, and provide a comparative description of the physical processes represented by each model.

The first step was to evaluate and compare the differences in modelling results from the DWF, SSF and SSF-DWF models. The evaluation focused on the performance of the models in reproducing the hydrographs in overall terms, but also, more specifically, on their ability to reproduce the characteristic stages of floods: rising flood waters, high discharges, and flood recession. These stages were defined as follows:

– Rising flood waters: the period between the moment when the observed flowrate exceeded the mean inter-annual discharge of the catchment and the date of the first flood peak.

– High discharges: this stage includes the points for which the observed flow was greater than 0.25 times the maximum flow during the event.

– Flood recession: this stage begins after a period of $t_c$ (the catchment concentration time according to Bransby's formula (Pilgrim and Cordery, 1992): $t_c = 21.3 \cdot L / (A^{0.1} \cdot S^{0.2})$) after the peak of the flood, and ends when discharge is rising again (or, where appropriate, at the end of the event - the time of peak flooding + 48h).

The Qmed_INT [%] score was used to evaluate the ability of the models to reproduce overall flows, rising flood waters and high discharges. For the time interval considered, Qmed_INT defines the percentage of points within the modelling acceptability zone for the median forecast of the calibrated model, with the acceptability zone determined by $\Sigma_{mod}$ et $\Sigma_{\hat{y}}$. Conversely, Qmed_INT was not used in the evaluation of the capacity to reproduce recessions, because the calculation of this score during the recession interval strongly depends on performance at high discharges. We therefore choose to make a visual comparison of the simulated and observed recession curves, $Q(t) = f\left(log\left(-\frac{dQ(t)}{dt}\right)\right)$, which are characteristic of a catchment's hydraulic discharge properties (Troch et al., 2013; Kirchner, 2009). Lastly, the evaluation was completed by a description of the a priori and a posteriori modelling errors in order to identify those that were inherent in the choice of model structure, regardless of the calibration strategy adopted.

A second part of the work was to study the flow processes generated - surface and subsurface flows, and flows at depth, in order to: i) identify the impact of the choice of a model structure on the properties of the simulated hydrograph, ii) identify the dominant processes for each catchment, and iii) assess the fit of modelling results to the known hydrological behaviours (cf. Section 1.3).

Lastly, the calibration strategy meant that it was not possible to determine a unique suitable model structure for some catchments. To illustrate this, we considered in detail four "model + parameter set" configurations that were all equally plausible in terms of describing an integrated hydrological response in order to clarify the actual differences induced by the modelled processes and identify options to better configure the models.





## 5 Results

### 5.1 Performance of the models

Figure 8 shows the Qmed_INT scores obtained after calibration of the DWF, SSF and SSF-DWF models for each catchment studied. It also shows the mean and standard deviations obtained from the series of calibration (top) and validation (bottom)
events, calculated from all or parts of the hydrographs.

#### 5.1.1 Assessment of performance by catchment

This section analyses the differences in performance, depending on the model used and the catchment studied. The DWF model assuming deep infiltration and the formation of an aquifer flow in altered bedrocks showed better performance in the Ardèche catchment (♯1), while in the Gardon (♯2) and the Salz (♯4) catchments, the SSF and SSF-DWF models, assuming the
formation of subsurface flows due to the activation of preferential flowpaths (SSF), local saturation and development of flow at depth (SSF-DWF), produced the most accurate results. On the Hérault catchment (♯3), the modelling results obtained with each model, in terms of Qmed_INT, were less obvious, although the SSF-DWF model seemed to stand out to some extent. The differences in model performance were more pronounced for the validation events. The better-performing models tended to be more consistent, with equivalent Qmed_INT scores on calibration and validation events (for example, the DWF model on the
Ardèche (♯1) or the SSF and SSF-DWF models on the Gardon (♯2)). There was also a deterioration in performance in several models that had already been judged less effective (for example, the SSF and SSF-DWF models on the Ardèche (♯1), or the SSF model on the two catchments of the Hérault, ♯3c and ♯3d).

#### 5.1.2 SSF compared with SSF-DWF

As a reminder, the difference between the SSF and SSF-DWF models is that the latter has an extra calibration parameter -
$C_{kdw}$ - to be able to initialise a significant lateral flow in the subsoil horizons of the soil column (see Equation 3). The lateral hydraulic conductivity in the deep layer is configured using the hydraulic conductivity from BD-sol: $K_{dw} = C_{kdw} \cdot K_{BDsol}$, with $C_{kdw}$ set to $0.02 \cdot C_{kss}$ in the SSF model and calibrated in the SSF-DWF model. The small differences between the SSF and SSF-DWF models showed that this flexibility does not produce any significant improvement, with the exceptions of the Ardèche catchment at Meyras and the Hérault catchment at Valleraugue. These two areas have a number of common features
that could explain the similar modelling results: they are at the heads of high elevation catchments with steep slopes (Table 1), and are subject to considerable annual meteorological forcing. Therefore calibration of the saturation hydraulic conductivity parameter of the subsoil horizon tended to result in a significant flow at depth for these two catchments ($\frac{1}{C_{kdw}} \in [3, 36]$ for ♯1d and $\frac{1}{C_{kdw}} \in [5, 34]$ for ♯3d, Figure 9, with this ratio set to 50 in the SSF model). In general, the calibration of the $C_{kdw}$ parameter of the SSF-DWF model (Figure 9) seems to be correlated with the more or less sustained, annual hydrological
activity of the catchments: the confidence interval of the $C_{kdw}$ coefficient is restricted to low values for the catchments with low mean inter-annual discharges (Figure 9, ♯2a, ♯2b, ♯2c, ♯3a, ♯3b, ♯4) and inversely for ♯1, ♯3c and ♯3d.





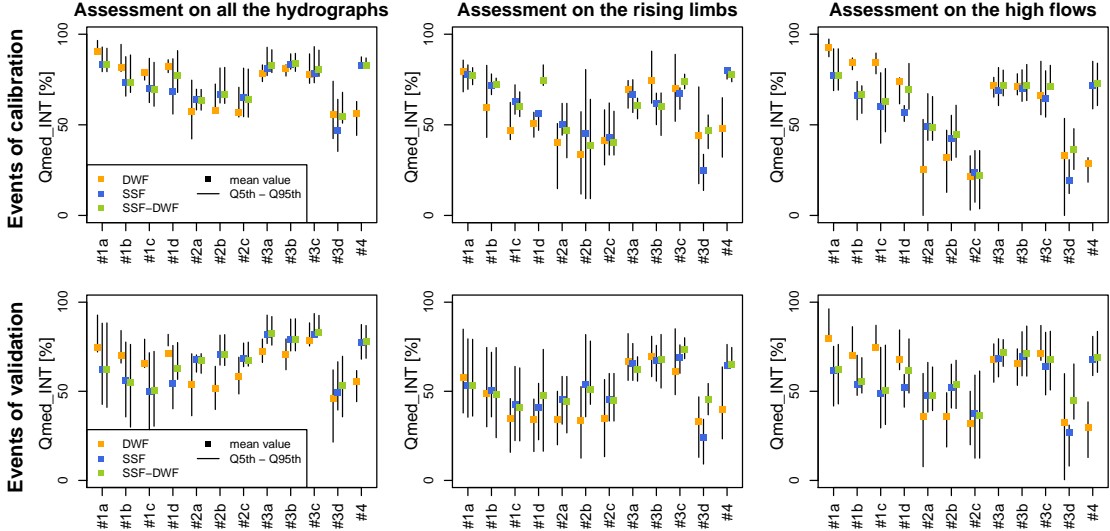

**Figure 8.** Qmed_INT scores: mean Qmed_INT scores obtained for the calibration (top) and validation (bottom) events, by model and catchment. The Qmed_INT scores were calculated for the whole hydrograph (left), modelling of the rising flood waters (centre), and modelling of high discharges (right).

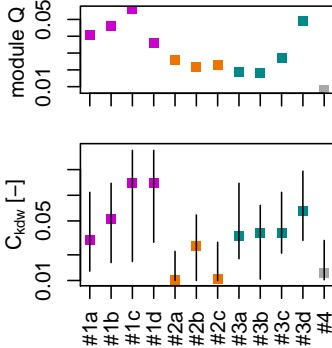

**Figure 9.** Top: Mean inter-annual discharge ($m^3.km^{-2}.s^{-1}$) for the catchments. Bottom: a posteriori distribution of the calibration of the subsoil horizon hydraulic conductivity in the SSF-DWF model (the $C_{kdw}$ parameter, Equation 3)

### 5.1.3  Representation of rising flood waters and high-volume discharges

Considering detailed results for all periods covered by hydrographs, the SSF and SSF-DWF models on the Gardon and the Salz catchments produced the most uniform results, since both the simulations of rising flood waters and high-volume discharges demonstrated the superior performance of these models. The results for the Ardèche were not as clear (Fig. 8, (♯1)), because we observed that the DWF model produces the best simulation of high-volume flows. Conversely, the DWF model deals slightly less well, overall, with rising flood waters. All the models tend to underestimate initial flows prior to the event and during the





onset of a flood. The DWF model, in particular, exhibits this modelling weakness (see, for example, the onset of floods in the hydrographs for the 18/10/2006 and 01/11/2014 events in Ucel (♯1b), Figure 11), which explains the poorer performance. On the Hérault, a detailed evaluation enabled us to compare the performance of the different models. On the one hand, the DWF model shows a more mixed performance for rising flood waters, reflecting a wider Qmed_INT confidence interval, which

5    indicates greater uncertainty in forecasting the timing of rising flood waters. In addition, this model performed the best on the Hérault catchments at Laroque (♯3a) and Saint Laurent le Minier (♯3b); while the SSF-DWF model generated the best results for the upstream catchments of La Terrisse (♯3c) and Valleraugue (♯3d). These results suggest a marked influence of the physiographic properties on the development of flow processes because they are correlated with the differences in the geological and topographical properties of the Hérault (♯3; see Figure 3 and Table 1). The hydrological behaviours simulated

10   for the Valleraugue and La Terrisse sub-catchments, which are predominantly granitic and schistose, and where slopes are very steep, can be distinguished from those of Laroque and Saint-Laurent-le-Minier, which are mainly sedimentary and in the form of large plateaus.

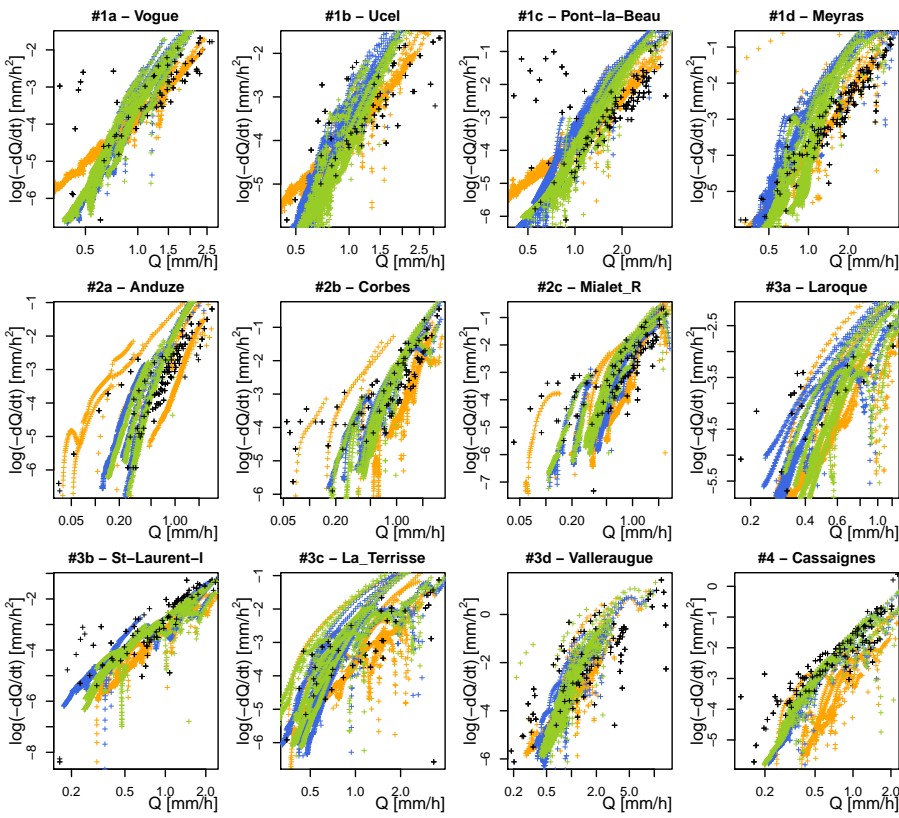

**Figure 10.** Comparison of the modelled and observed characteristics of flood recession. Black: flood recession points for observed flows; orange: flood recession points for flows modelled using the DWF model; blue: flood recession points for flows modelled with the SSF model; green: flood recession points for flows modelled with the SSF-DWF model.





### 5.1.4 Representation of flood recessions

Visual inspection of the hydrographs showed that some models produced a better fit on certain catchments. An example of this was the DWF model on the Ardèche catchments (for example, the simulation of hydrographs at Ucel; ♯1d, Figure 11), which provided a much better fit for flood recession. The DWF model more accurately simulated the slow flood recession in the Ardèche catchment.

The flood recession characteristics, $Q(t) = f(log(-\frac{dQ(t)}{dt}))$, reflected the catchment's release properties. Figure 10 compares the simulated and observed flood-recession curves for each catchment. The catchments can be divided into three groups. For the Ardèche catchments (♯1a, 1b, 1c and 1d), the DWF model is considerably more accurate in reproducing flood recession, especially at the moment when the waters begin to recede. For the Salz-Cassaignes (♯4) and Gardon-Anduze and Corbes (♯2a, ♯2b) catchments, the SSF-DWF and SSF models performed better in reproducing recession curves. Conversely, for the other catchments, there are no distinctions to be drawn on how realistic the models'results are for this criterion, as can be seen for the Hérault catchment at Valleraugue (♯3d). For this third group, either there is no clear hydrological signature of the observed recessions (♯3a, ♯3c), or the characteristic recessions predicted by the models cannot be distinguished (♯2c, ♯3b).

### 5.1.5 Modelling errors inherent in the models' structures:

Figure 11 shows an illustration of the simulation results of the three models using an example catchment (the Ardèche catchment at Ucel, ♯1b). It shows the simulated hydrographs, and their confidence intervals, compared with observed flows, as well as the inherent errors in the simulations. This highlights the modelling errors due to the choice of model structure (DWF, SSF or SSF-DWF models).

Representing the soil column with either one compartment (the DWF model) or two compartments (SSF or SSF-DWF models) leads to distinct (grey) a priori modelling errors in the variation intervals. The first structure (the DWF model) constrains the simulated flows at the beginning of the event, before the onset of precipitation, because the variation interval of the modelling errors is low at that point. More specifically, it tends to underestimate the initialisation discharges because the variation interval of the errors over this period is predominantly negative. This may explain this model's relative difficulty in reproducing the onset of floods, since the calibration of the parameters did not allow the acceptability zone on this part of the hydrograph to be reached.

Likewise, it can be noted that the one-compartment structure (in the DWF model) allows flexibility in the modelling of high discharges and flood recessions, because the variation intervals in the modelling errors was quite large over these periods in the hydrograph. However, it also led to the underestimation of high discharges and flood recessions. In fact, the modelling error interval had a negative bias with respect to the acceptability zone. The calibration finally allows the simulations to be selected, at the intersection of the acceptability zones and a priori confidence in modelling errors. This generally corresponds to the calibration of a low-depth altered rock $D_{WB}$, in order to make the model more sensitive to soil saturation and more responsive, via the generation of early runoff.





Conversely, the two-compartment structure (the SSF and SSF-DWF models) offers flexibility in modelling the beginning of events, flood warnings and high discharges, but the ability to model flood recessions is more constrained. The relative position of the modelling-error confidence interval, with respect to the acceptability zone, showed that the structure leads to an unbiased estimate of the onset of a flood, a slight overestimation of high discharges, and an underestimation of flood recessions.

In the SSF and SSF-DWF models, the addition of a flux calibration parameter in the subsoil horizons, not surprisingly, led to wider variations in the a priori modelling errors. A surprising finding, however, was that the calibration of the lateral conductivity of the deep layer, $C_{kdw}$, seemed to affect only the simulation at the beginning of the hydrographs (the events of 01/11/2011 and 13/11/2014), and had very little effect on flood recessions. This last point was also visible in the analysis of flood recessions, where we observed a high degree of similarity in the flood recessions simulated by the SSF and SSF-DWF

models, whatever catchment was being studied. The calibration of $C_{kdw}$, in fact, only influences the support volume from the subsoil horizons, but not the rate of decline (and, hence, not the recession). This is determined by the exponential term in equations 1, 2 and 3). It appears that the limited variations in the speeds of flood recession for the SSF and SSF-DWF models can be explained by the value of $m_\theta$ in Equation 2, which determines the rate of decay for most of the runoff as a flood recedes.

### 5.1.6   Interpretation of the hydrological functioning of the catchments studied

On the basis of the calibration and performance of the DWF, SSF and SSF-DWF models, the catchments can be divided into several groups:

- The SSF and SSF-DWF models showed better overall performance (with no particular pattern) on the Gardon (♯2) and Salz (♯4) catchments. This suggests, on the one hand, rapid catchment reactivity, and on the other, formation of the flows in the soil through local saturation tied to the climate forcing. The contrasting physiographic characteristics of these

catchments suggest that there are different explanations for this better fit of the SSF-DWF model. On the Gardon, the very high intensities of the observed events (Table 2) and/or the low soil depth (Table 1) may explain the limitations on vertical infiltration due to the properties of the soil and/or geological bedrock; as a result, the rapid formation of a saturated zone at the top of the soil column, favours runoff and subsurface flux by activating preferential paths in the soil. On the Salz (♯4), the soil is deeper and the precipitation intensities lower. On the other hand, the geological bedrock

composed of marls, sandstone and limestone is assumed to have low permeability and the soil is less conductive due to its predominantly silt-loam texture. As a result, despite the lower forcing intensities, the surface soil can reach saturation, which explains why the SSF model offers the best fit.

- The considerable hydrological responses, in terms of volumes, on the Ardèche, appear to be linked to hydrological activity at depth, including that taking place during intense floods, as suggested by the better fit of the DWF model.

Here, in particular, the model gives a better representation of the relatively slow and uniform hydrological recessions from one event to the next, reflecting an aquifer-type flow whose discharge properties are governed by the properties of the catchment bedrock only. The somewhat delayed flood timing that the structure of the one-compartment model imposes seems to indicate that there are more rapid flows at the beginning of an event, which this model structure was



**Figure 11.** Calibration of the three models for the Ardèche catchment at Ucel, ♯1b). The results of the simulation of five flood hydrographs, and the inherent modelling errors for each model (top: DWF; centre: SSF; bottom: SSF-DWF). The median simulation and the simulation confidence interval are shown, respectively, in red and salmon. The confidence intervals of the measured flows and the acceptability zone are shown, respectively, in green and blue. The a priori modelling errors for each model (i.e. with no calibration) are shown in grey.




not able to represent. An initial explanation for this may lie in the design of the model: the drainage network being structured into 1 km² drained areas. The comparison with the observed hydrographic network for the catchment showed an under-representation of the upstream drainage network, which may have resulted in a delay in the modelling of the signal, despite the model offering a good overall fit. A second possible explanation is the default calibration, which uses a uniform depth of active subsoil horizons, $D_{WB}$, during a flood. This might mask the appearance of local saturation zones, and the subsequent runoff due to shallow soil and discontinuities in the permeable base layer (for example, in the downstream sedimentary layers, where infiltration tests have shown the appearance of runoff, see Section 2.1). In contrast, the SSF and SSF-DWF models do not display this weakness because the varying nature of soil depths ($D_{BDsol}$, which determines the depth of the upper compartment) allows the rapid development of flows via preferential paths in the soil blocks, thus enabling the simulation of such local dynamics.

– At the catchment heads (♯1d, ♯3c, and ♯3d), we observed superior performance from the DWF and SSF-DWF models, with a particular improvement in the forecasting of rising flood waters when using the SSF-DWF model. This suggests the presence of several types of flow in the soil with strong support from flows at depth, which corroborates the high mean inter-annual discharges associated with these catchments, and additionally the presence of rapidly formed flows, providing a good simulation of the rising flood waters. It should be noted that, here again, modelling the drainage network for an area greater than that observed on these steep-sloped catchments can also affect the results.

– On the Hérault, in spite of the fairly similar results generated by the models, we observed differences in model performance at the catchment heads (♯3c, ♯3d), where the SSF-DWF model performed significantly better, and at the downstream catchments (♯3a, ♯3b) where the DWF model performed better - for these catchments only - in simulating rising flood waters. An interpretation of hydrological functioning is nevertheless not possible, given the good overall results offered by other models and that no distinctions can be drawn according to other criteria, such as performance in terms of the simulation of flood recession, for example.

## 5.2 Characterisation of the hydrological processes simulated

Each time a model is run it generates its own paths for water flow as it attempts to reproduce the hydrograph. Figure 12 shows the proportional volumes of the water making up the hydrographs, that arise from catchment runoff which has not passed through the soil at any point. We can distinguish the contributions of these surface flows on the whole of the hydrograph (Figure 12, left) and those that support high discharges (Figure 12, right).

We can note that the SSF and SSF-DWF models suggest very similar proportions of subsurface flows, including those at the catchment heads. Calibration of hydraulic properties at depth influences - as intended - only the proportions of subsurface and deep aquifer flows (which are not shown here).

The DWF model suggests a larger contribution from runoff to the generation of high discharges, whatever the catchment modelled. In fact, we observed a 15 to 30 % increase in the proportions of surface flow between the DWF model and the SSF and SSF-DWF models. The performance of the DWF model was noticeably different on the Gardon (♯2) and Salz (♯4)





catchments, where simulated runoff was much more pronounced over the entire hydrograph. Here, the DWF model showed runoff proportions ranging from 40 to 98 %. However, the few experimental measurements made on the Gardon (Bouvier et al., 2017; Braud et al., 2016) provide firm evidence on the proportions of new water - i.e. water resulting from meteorological forcing during the event - which range from 20 to 40 % of the volumes in the hydrograph. This clearly points to a lower

runoff rate. Even though these experimental results only represent activity in the granitic part of the catchment, they appear to call into question the hydrological functioning suggested by the DWF model. Conversely, the observations lend support to the results obtained by the SSF and SSF-DWF models, where runoff rates were between 19 and 62 %. On the Salz there are no experimental observations available, and, therefore, observed results cannot be corroborated, by the orders of magnitude of the simulated surface flows. Nevertheless, in view of the performance of the different models, the SSF and SSF-DWF model

structures appear to be more pertinent for characterising the types of processes occurring. Taking the most suitable models for the catchments studied, an estimate of the degree of contribution of surface flows to the hydrographs can be made: (i) between 4 and 31 % for the main Ardèche catchments (♯1a, ♯1b, ♯1c), according to the DWF model, and between 0 and 40 %, or 10 and 29 %, on the Ardèche catchment at Meyras (♯1d), according to the DWF and SSF-DWF models, respectively; (ii) between 17 and 53 % (62 %) on the Gardon (♯2a, ♯2b, ♯2c) catchments; (iii) between 11 and 31 % in the Salz catchment (♯4); iv) between

5 and 58 % on ♯3a and ♯3c, 15 and 63 % on ♯3b, and 5 and 34 % on ♯3d according to the DWF model, or between 10 and 43 % on ♯3a and ♯3c, 11 and 58 % on ♯3b, and 4 and 20 % on ♯3d according to the SSF-DWF model.

  The above uncertainties are related to the parameterization of the models, a consequence of the equifinality of the solutions when calibrating a hydrological model against the sole criterion of the reproduction of the hydrological signal. While, in terms of plausibility, several sets of parameters may be equivalent, even for the same model, these sets of parameters are likely to

lead to different hydrological functioning. This is especially the case for the DWF model, for which the relative proportions of processes simulated depend on the choice of $D_{WB}$. Conversely, on the downstream catchments of the Hérault (♯3a, ♯3b), it can be noted that the variation intervals of the surface flows estimated by the three models overlap. On the one hand, this suggests that the models generate quite similar estimates of the proportions of surface flows. On the other, it may explain why the three models can achieve good reproductions of the hydrological signal - in that the calibration step makes it possible, from

an integration point of view - to obtain an analogous distribution of the flows.

  Analysis of the distribution of the flows between those passing through the soil and those flowing on the surface lends support to the SSF and SSF-DWF models being realistic for the Gardon (♯2) and Salz (♯4) catchments. However, drawing distinctions between the models through such an integrated description of processes is limited by the equifinality of the solutions. In order to better understand the different hypotheses on which the models are based, and the various likely parameters in the hydrological

changes that take place in the catchments, other variables, such as (spatialised and integrated) changes in moisture levels in the catchments or the flow velocities generated by modelling choices, have to be considered.

  Next, we describe the detailed results of four simulations, also considered to be plausible according to the DEC criterion, obtained from the DWF and SSF models as well as four sets of parameters (see Table 3). We considered the Hérault catchment at Saint-Laurent-le-Minier ((♯3b) because the criteria previously used had not shown any one model to be more representative.




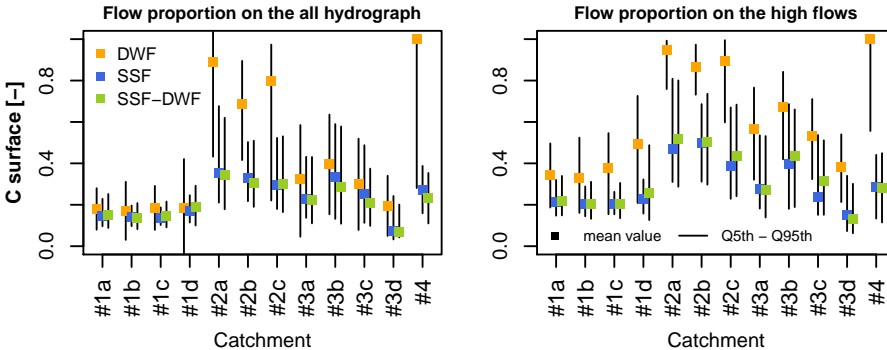

**Figure 12.** Proportion of surface runoff in the flows at the outlet. Left: The proportion over the whole hydrograph; right: the proportion at high discharges (Observed flow greater than 0.25 times the maximum flow during the event).

The objective was to highlight how the models differed in terms of flow development, and what compensations occurred between processes to allow the equifinality of solutions.

## 5.3 Detailed study of four plausible simulations on the Hérault watershed at Saint Laurent-le-Minier

**Table 3.** Realistic models and parameter sets for the Hérault catchment at Saint-Laurent-le-Minier (♯3b). $C_{soil}$: the contribution to the hydrograph of flows passing through the soil ; $C_{kdw}/C_{kss}^*$ : the value of the parameter $C_{kdw}$ for model DWF (Equation 1) or the value of the parameter $C_{kss}^*$ for the model SSF (Equation 2).

| ID | NSE | $D_{WB}[m]$ | $C_k[-]$ | $C_{kdw}/C_{kss}^*[-]$ | $n_r[-]$ | $n_p[-]$ | $C_{soil}[\%]$ |
|------|------|------|-------|-------|--------|--------|------|
| DWF1 | 0.82 | 0.15 | 17.3 | 8711 | 19.6 | 19.11 | 61 |
| DWF2 | 0.84 | 0.11 | 2.34 | 4416 | 19.16 | 7.63 | 39 |
| SSF1 | 0.89 | 0.40 | 15.81 | 45284 | 15.96 | 5.86 | 68 |
| SSF2 | 0.89 | 0.34 | 2.08 | 22543 | 14.06 | 6.42 | 53 |

The figure 13 compares the changes over time in the state of soil saturation and the different simulated flow velocities of
5    four "model + parameter set" configurations based on the DEC score (Table 3). Figure 14 compares the spatial distributions of these variables, at a given moment, as an example. In terms of hydrographs, quite logically given the similar likelihood scores, the simulations differed very little. Overall, the DWF1 configuration anticipated flood peaks; the DWF model (in the DWF1 and DWF2 configurations) generated greater flows at the end of rain episodes; and these same configurations resulted in a slight underestimation of peaks for floods of average intensity (18/10/2009 and 05/03/2013) and, conversely, an overestimation
10   of the peaks for exceptional floods (12/03/2011 and 01/11/2011), compared with the SSF model (in the SSF1 and SSF2 configurations).





The notable difference in the generation of hydrographs was the contribution of the different simulated flowpaths. The proportions of water passing through the soil column (via sub or surface-soil horizons) were highly variable: with an average of 39 % for the DWF2 model, 53 % for the SSF2 model, 61 % for the DWF1 model and 68 % for the SSF1 model. This is due both to the structural choices (DWF and SSF) which involve a saturation dynamic and the incorporation of different types of

flow, and the choice of the parameters which involves flow velocities of differing orders of magnitude.

Figure 13-b) shows the different saturation dynamics involved in the DWF and SSF structures. The DWF structure entails continuous drainage of the catchment, including at initialisation. This resulted in a noticeable decrease in the water content of the soil at the beginning of an event, which slows down saturation during the onset of flooding. The overall discharges (solid line) from the catchment simulated by the SSF model were distinguished by a gradual decrease towards a state of equilibrium,

as opposed to the DWF model, for which the decay was faster. Taking Figure 14 (the left-hand column), we can also observe differences in spatial dynamics. The DWF model produced a greater contrast in saturation levels between different areas of the catchment. This results from the decrease in the simulated flows as a function of water height (cf. Section 3.2, Equation 1), which makes the draining of each grid cell sensitive to uniform soil depths. With the SSF model, the overall catchment saturation levels appear to be more related to the topography, and less related to the rain episode: we observed saturation of the

cells close to the drainage network, and, conversely, lower water content in the upper reaches of the catchments. In fact, for the SSF model, rainfall forcing is mainly involved in saturation of the upper soil layer (the dashed lines in Figure 13-b), which reacts very rapidly to precipitation.

The flow velocities simulated in the soil (Figure 13-c) were linked to the saturation dynamics. At the start of flooding, the SSF structure resulted in an early increase in flow velocities due to a higher saturation level of the upper soil layer. Moreover,

the flow model chosen (Equation 2 and fixing of the parameter $m_\theta$) to simulate the activation of preferential paths in the SSF model allowed a much greater variation of simulated velocities over the short period during which the watershed was saturated. Conversely, for the DWF model, the variation interval of simulated velocities is two to four times lower, and the reaction to changes in soil moisture appears to be more linear. The choice of parameters - in particular $C_{kss}$, here - influenced the order of magnitude of the simulated velocities but not the evolution over time, which depends on the structure of the model (the flow

modelling equation and the representation of one or two compartments).

The spatial distributions of the flow velocities in the soil (Figure 14, centre) showed similarities with the areas affected by the flows. For the four configurations, the development of flows in the soil only partially reflected the state of saturation, but it was correlated with the physiographic properties of the soil (topography and thickness) and the spatial distribution of meteorological forcing. The different orders of magnitude in the simulated velocities reflect the calibrations of the $C_{kss}/C_{kdw}$

parameters in the four configurations.

The evolution of runoff velocities in the catchment area (Figure 13-d) showed that the main difference between the "model + parameter set" configurations is the orders of magnitude of the runoff velocities. This is mainly due to the number of grid cells in the catchment with excess infiltration, and reflects the influence of the infiltration control parameter, $C_k$, and the depth of the subsoil horizon, $D_{WB}$. We also noted, at the end of the event, the presence of average, non-zero runoff rates on the catchments

with the DWF model, a consequence of grid cells that are still saturated.





The spatial distributions of the flow velocities on the catchments (Figure 14, right) show the two types of functioning suggested by the four proposed configurations. Either the runoff was generated by exceeding the storage capacity of the soil; this was the case for configurations DWF1 and SSF1, where the grid cells with non-zero runoff velocities corresponded to the grid cells where the saturation state of the soil column had been reached, or, runoff was generated by exceeding the infiltration capacity of the soil; this was the case for configurations DWF2 and SSF2 for which the coefficient $C_k$, set at a low value (cf. Table 3), limited infiltration.

The changes in runoff velocities in the drainage network (Figure 13-e) reflected the soil saturation dynamics (Figure 13-b). For the SSF model, an early increase in velocities in the drainage network was observed; this is due to the fast saturation of the upper compartment of the soil column, producing consequently interflows through activation of preferential flow paths at the beginning of the event. The DWF model yielded a more contrasting variation in the runoff velocities in the drainage network, mirroring variations in soil saturation levels. Finally, it can again be noted that only the structure of the model influenced the evolution over time of flows in the drainage network, whereas the choice of parameters - particularly, here, $n_r$ and $n_p$ - affected the order of magnitude of the simulated velocities. Taking the four configurations, the selection of plausible parameter sets appears to show a correlation between the parameters $C_k$, and $n_r$ and $n_p$. This was already emphasized by the high values of the Pearson correlation coefficient, especially for the Gardon catchment at Anduze (♯2a): $\rho_{DWF} = 0.46$ and $\rho_{SSF} = 0.18$. This shows the necessity of slowing down flows in the drainage network when a larger proportion of runoff from the catchments is simulated (i.e $C_k$ is low). In all cases, where the values of $C_k$ are low, the transit flows through the ground were also slower (i.e. the values of $C_{kss}$, $C_{kdw}$ were low). Thus, as a result of the model calibration, a degree of compensation occurs in the simulated transfer times between the various water paths, from the hillslopes to the drainage network, and from the drainage network towards the outlet.

## 5.4 Discussion

In this comparison of the simulated processes, the equivalence of the configurations, presented in terms of integrated modelling of the flow at the outlet, and consequently in terms of likelihood, is refuted by the differences generated in:

– the proportions of water passing through the ground or over the surface, linked to the infiltration control mechanism, governed here by the chosen configuration.

– the saturation dynamics of the soil, which are linked to the flows developed in the soil, governed here by the structure of the chosen model.

– the orders of magnitude of the simulated flow velocities, which are related to both the choice of the model structure and the parameterization.

A detailed description of the configurations, together with an estimation of the structural errors in the modelling, allows better visualisation of what the different hypotheses of hydrological functioning involve, and points to new options for assessing models, as well as the potential contributions from new knowledge/observations:





**Figure 13.** Comparison of the results of four equally plausible simulations on the Hérault at Saint Laurent le Minier (Table 3). a) Flood hydrographs (solid lines) and outlet flows transiting via the soil (dashed lines). b) Evolution in the overall moisture content of the soil column. c) Evolution in simulated mean velocities in the subsoil horizon (DWF model) and in the upper part of the soil column (SSF model). d) Average runoff velocities on the hillslopes. e) Average runoff velocities in the drainage network.

  – The DWF and SSF structures generated vertical dynamics and distinct spatial saturation patterns. The current availability of high-resolution telemetry measurements with high spatial coverage (for example, Sentinel-1-based satellite Earth



**Figure 14.** Spatialised outputs for a given moment during the event of 18/10/2009 (during the development of the flood, where $Q = 74\ m^3.s^{-1}$): a-d-g-j) soil moisture conditions simulated, respectively, by the configurations DWF1, DWF2, SSF1, SSF2; b-e-h-k) discharges in the soil simulated, respectively, by the configurations DWF1, DWF2, SSF1, SSF2 (N.B: different colour scheme); c-f-i-l) surface flow velocities simulated, respectively, by the configurations DWF1, DWF2, SSF1, SSF2.

Observation data (Enenkel et al., 2016; Cenci et al., 2017)) offers the opportunity to conduct a qualitative assessment of soil moisture patterns. The temporal resolution (up to six days) is not adapted to flash-flood time scales and prevents their use for real-time evaluation of hydrological simulations. However, observing some saturation patterns for a number





of events during, or shortly after, an episode would provide an interesting research avenue, in terms of distinguishing the hydrological reactions of the catchments in a spatialised manner, which could help confirm the accuracy of the models tested.

– The different flow proportions related to the structure of the model selected (use of the DWF model tends to result in more runoff on slopes) and its calibration emerge as new objectives for constraints, because they imply distinct hydrological behaviours. Tracing flows via isotopic measurements is not suited to the meso-scale catchments studied, nor to the spatial representation of the MARINE model, which assumes an instantaneous and complete mixing of the water volume and does not calculate residence times (McDonnell and Beven, 2014). Conversely, the use of an indicator of the presence of runoff, such as diatom tracing (Pfister et al., 2017b), suspended particles or the turbidity of water, offers an indirect means of detecting the degree of surface flows in a flood, and could make it possible to better constrain the partitioning of the hydrographs.

– The different proportions in the simulated flows are allowed by the simulation of transfer times, of varying length, on the different water paths: runoff, flows through the soil, and via the drainage network. These arise, in particular, as a result of the calibration of flow velocities having different orders of magnitude. It would be difficult to envisage a constraint on the orders of magnitude of the simulated velocities because the scale of modelling (where, as a reminder, $dx\ 100\ m$) encompasses macrostructures (for example, preferential paths) that cannot be quantified without detailed analysis. Conversely, separate optimisation of the drainage network, and the parameters that control flow on the hillslopes, would limit the possible compensations between the transfer times modelled. In particular, intermediate hydrometric stations could be used to calibrate the transfer function of the hydrological signal from the drainage network to the outlet.

– Finally, the evaluation based on the ability of the simulations to reproduce the characteristic stages of floods demonstrates the greater impact of the choice of model structure on the rise and recession of floods. They, therefore, point the way towards an optimal consideration of these parts of the hydrograph. The choice of an evaluation score based on a comparison of time series proved pertinent as a result of its sensitivity at the onset of a flood. The uncertainty in flow measurements was systematically taken into account for all catchments. In order to refine the information on the measured flows (and, more precisely, the average range of flows for rising and receding floods), it would make sense to carry out specific calculations for each hydrometric station and its associated discharge curves (using the Baratin Method (Le Coz et al., 2014), for example).

## 6  Conclusions and Perspectives

### 6.1  Summary of the study's objectives and methodology

The objective of the study was to improve our understanding of flash flooding on the French Mediterranean Arc. In particular, attention was paid to the dynamics of soil saturation in catchments during these events, and their possible relationship with



the physiographic diversity encountered. The method used consisted in considering hydrological models as a diagnostic tool to test hypotheses about the functioning of catchments.

Based on the structure of the MARINE model - a hydrological model with a physical and distributed basis - three types of dynamic of soil saturation were postulated and tested. In the first case (the DWF model), we assumed an aquifer dynamic, with

infiltration at depth, and the generation of strong base support, according to the volume of infiltrated water; in the second case (the SSF model), it was the activation of preferential paths at the soil/altered rock interface that generated the majority of the flows passing through the soil, with the lower part of the soil column serving only as a storage reservoir; and in the third case (the SSF-DWF model), there was flow generation via both the activation of preferential pathways, initially by saturation of the top of the soil column, and a significant increase in the base flux via the subsequent infiltration of water present at deeper

levels.

The same calibration strategy was used for the three models on a set of 12 catchments which are representative of the diverse characteristics of the Mediterranean Arc. Whether a model offers a good fit was evaluated on the basis of: scores representing overall, or partial model performance in terms of simulating the hydrographs; the proportions of the processes simulated; and the timing and form of flood recession.

## 15  6.2   Conclusions on our understanding of the processes involved

From the application and validation of the three hydrological models, the 12 catchments of the study could be classified into four categories: i) the Gardon and Salz catchments, for which the SSF model was better suited to reproducing the hydrological signal. For these catchments, this highlights the importance of local and surface soil dynamics in the generation of flows, especially at the beginning of a flood; (ii) the Ardèche catchments, for which the DWF model most accurately reproduced the

observed flows. This demonstrates more regular and integrated hydrological functioning at the catchment level, with the flows generated being directly related to the moisture history and rainfall volumes; (iii) the Hérault catchments at Valleraugue and La Terrisse, and the Ardèche catchment at Meyras, which have steep-sloped catchment heads, where the SSF-DWF model stood out, suggesting both sustained and significant hydrological activity at depth during flash floods, and surface activity in the establishment of early flows at the beginning of events; (iv) the Hérault catchments at Laroque and Saint-Laurent-le-Minier,

for which no model showed any significant difference.

For each catchment, the best performing models were those where results reflected the available knowledge and observations on the overall hydrological functioning of the catchments, and where estimates of the different flow processes corresponded to experimental observations. The results suggest that the behaviour of catchments under extreme forcing is a continuation of the hydrological functioning normally encountered. Several earlier studies have pointed to a potential correspondence between

hydrological functioning and the nature of the geological bedrock. This is in evidence on the Hérault, where the evaluation of the three models highlighted different hydrological behaviours which are linked to differences in the geological nature of the catchments. On the other hand, the Gardon and Ardèche catchments, both of which have mainly granitic and schistose geology, exhibited different behaviours, with that of the Gardon being comparable to the hydrological behaviour of the sedimentary catchment of the Salz. These results, however, did not contradict the earlier studies. These studies were suggesting a possible





relationship between storage capacity in the substratum and the nature of the geological bedrock, whereas the differences highlighted here concern the formation of flows in the soil.

Lastly, identifying the most pertinent hydrological models for each catchment enables the key elements in the generation of flash floods to be highlighted, which, in turn, could serve to further develop methods for forecasting flash floods. For example,

5 distinctions in hydrological behaviour revealed between the catchments of the Gardon and the Ardèche may explain that taking into account the spatial nature of precipitation in a flash flood forecasting method results in an improvement only on the Gardon and not on the Ardèche (Douinot et al., 2016). Indeed, in the present study, the Gardon catchment appears to be more sensitive to the local dynamic of the soil water content than the Ardèche one, corroborating the sensitivity to spatial distribution of the rainfall revealed in Douinot et al. (2016).

## 10   6.3   Conclusions about the method used

The use of the hydrological model as a diagnostic tool allowed the classification of the catchments studied. It also contributes to the overall knowledge of these catchments in order to improve understanding of hydrological functioning during flash floods. The study also demonstrates: i) the complementarity of field observations in the interpretation of results, ii) the limitations in the evaluation and drawing of distinctions between models when constrained solely on the basis of the reproduction of an

15 integrated response; and (iii) the contribution that an analysis of the equifinality and model functioning can make to guide the choice of new and better constraints, and the strategic observations that need to be made in order to differentiate between equally plausible models. Lastly, distinguishing between models based on the evolution of internal variables - flow velocities and soil saturation states - makes it possible to highlight the value added by the descriptive potential of a distributed model with a physical basis, such as MARINE.

20 *Competing interests.*  The authors declare that they have no conflict of interest.

*Acknowledgements.*  This work was partly funded by the Eurorégion Pyrénées-Méditerranée (PGRI-EPM project) and the French central service for flood forecasting (SCHAPI).





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
