# Peer review of "Using a multi-hypothesis framework to improve the understanding of flow dynamics during flash floods"

_Hydrology and Earth System Sciences, 2017_

## Referee Comment (RC1) · Anonymous Referee #1 · 11 Jan 2018

This paper proposes a methodology for the analysis of catchment hydrological behaviors during flash floods, based on the introduction and comparison of several hypotheses in a distributed hydrological model. This topic is of broad interest for the hydrological scientific community, and is fully relevant in my opinion for a publication in HESS.

However, in its current form the paper suffers from a lack of detail and explanations on several aspects (calibration procedure, explanations related to some figures, ..), causing difficulties for a detailed understanding of the research content. The presentation of sections 4 and 5 should particularly be improved in my opinion (and maybe

organized in a slightly different way) to facilitate the overall understanding of the results and related analyses. The paper well illustrates the difficulties in the interpretation of modelling results, due to equifinality issues and lack of internal observations to confirm the nature of the main hydrological processes. Therefore, even if some solutions to cope with these difficulties are proposed here, I think the conclusions relative to the catchments behaviors (section 5.6) should finally be relativized and presented in the discussion section as the most reasonable assumptions, provided the modelling results obtained here.

**Specific comments**

- The abstract is very short and could be slightly more detailed.

- References would be welcome in section 1.2

- The description of the calibration procedure (section 4.1) and of the metrics for evaluation (section 4.2) are not sufficiently clear in my opinion, and should be improved:

- Please indicate how the "confidence intervals" are obtained for observations ($y_i \pm 2.\sigma_{y_i}$ ?) and also for modelling results. This should clarify why the uncertainties ranges mentioned in the text (respectively $20\%$ for observations and $10\%$ for modelling errors) are consistent with eq.(6) and eq.(7)

- please clarify the reason why the metric used for evaluation (QmedINT) is different from the one used for calibration (DEC) ?

- the definition of the "acceptability zone" should be provided ($y_i \pm 2.\sigma_{y_i} \pm 2.\sigma_{mod}$ ?)

- the "a priori" and "a posteriori" modelling errors are not defined. This clearly limits the interpretation of figure 11 (see hereafter).

- The presentation of modelling results (section 5.1) could also be enhanced:

- I think the separated presentation of each metric (overall hydrograph, rising limb, high-discharges, recession) does not help to give a synthetic overview of results. It seems that three main situations can be distinguished here: clear hierarchy (Gardons and Salz), contrasted hierarchy (Ardèche), or no clear hierarchy (equifinality, Hérault). These three situations could be illustrated based on a common analysis of all the metrics.

- Explanations in section 5.1.4 are poorly supported by figure 10 in my opinion. Please try to clarify this section and figure.

- The analysis proposed in section 5.1.5 is also difficult to follow based on figure 11, which does not well illustrate in my opinion the differences in models behaviors. Figure 11 indeed is difficult to understand: $\epsilon_{DDEC}$ is not defined, the definition of prior and posterior errors is again missing. Moreover, is not clear why the width of the acceptability zone does not vary with yi (not consistent with equation (7)). Please try to clarify this section and figure.

- Section 5.1.6: I think this interpretation on catchment behaviors arrives too early here. I think it would be better to put this in the discussion section, and to present these analyses as plausible assumptions, according to the modelling results.

- Section 5.2 may be renamed in a more explicit way, such as: "Analysis of relevance of the internal hydrological processes simulated". It could include both considerations on proportion of surface runoff (current section 5.2), and detailed analysis of velocities and water contents in the case of Hérault (current section 5.3)

- Conclusions: I don't' really agree with two of the conclusions:

- "For each catchment, the best performing models were those where results reflected the available knowledge and observations on the overall hydrological functioning of the catchments . . .". Actually, it seems that very limited information is
available on the real hydrological behavior, excepted maybe for the Gardon where detailed measurements were performed. Therefore, I would rather conclude that the modelling results help to draw consistent assumptions on hydrological behaviors, that can in some (rare) cases be confirmed by the existing knowledge and local observations.

- "distinction in hydrological revealed between the catchment of the Gardons and the Ardèche may explain that taking into account the spatial nature of precipitation in a flash flood forecasting method results in an improvement only on the Gardon and not on the Ardèche . . ." I think this conclusion is not really supported by the content of the paper. Moreover, another explanation could just be a difference in the rainfall spatial variability, which seems to be more pronounced in the Gardons catchment for climatic reasons.

**Technical corrections**

- Section 2.2:

  - The reference Ministère de l'Ecologie (2015) just corresponds to an URL, which could be adde directly in the text.

  - "These measurements were calibrated by forecaters at the French Flood Forecasting service by monitoring a network of rain gauges . . .". Sentence not clear, please reformulate.

- Figure 5: are $\theta_s$ and $\theta_i$ really the current and initial water contents respectively? Shouldn't rather $\theta_s$ be the saturation water content?

- Section 3.2, description of the modelling principles: the equations (1) to (4) and description of variables should be placed in the text with reference to figures 6 and 7

- Section 4.2: $\sigma_y$ and $\sigma_{mod}$ rather than $\Sigma_y$ and $\Sigma_{mod}$

- Section 5.1.5: "the variation interval of the modelling errors": I don't really understand, please define this.

- I finally suggest to check the overall quality of English.

---

## Short Comment (SC1) · 11 Feb 2018

We wish to thank the referee for their careful evaluation of the manuscript which would be hopefully more intelligible thanks to their constructive comments. Below are our detailed responses (in blue) to the comments (in black).

Sincerely yours,
* * *
[Figure]

*This paper proposes a methodology for the analysis of catchment hydrological behaviors during flash floods, based on the introduction and comparison of several hypotheses in a distributed hydrological model. This topic is of broad interest for the hydrological scientific community, and is fully relevant in my opinion for a publication in HESS. However, in its current form the paper suffers from a lack of detail and explanations on several aspects (calibration procedure, explanations related to some figures, ..), causing difficulties for a detailed understanding of the research content. The presentation of sections 4 and 5 should particularly be improved in my opinion (and maybe organized in a slightly different way) to facilitate the overall understanding of the results and related analyses. The paper well illustrates the difficulties in the interpretation of modelling results, due to equifinality issues and lack of internal observations to confirm the nature of the main hydrological processes. Therefore, even if some solutions to cope with these difficulties are proposed here, I think the conclusions relative to the catchments behaviors (section 5.6) should finally be relativized and presented in the discussion section as the most reasonable assumptions, provided the modelling results obtained here.*

We are grateful for the constructive comments. Most of them shift toward quite deep modifications of the sections 4 and 5. Following the reviewer's suggestions, the organization of section 5 has been changed, and many efforts were made to describe the methodology (section 4). The answers to the comments are listed below. The modifications made in the paper are quoted in italics. Line references correspond to the marked manuscript (attached file).
* * *
**Specific comments:**
The abstract is very short and could be slightly more detailed. The abstract has been reworded, giving now more details about the results (page 1, lines 5 - 10 :). Below, the reworded abstract :

*"A method of multiple working hypotheses was applied to a range of catchments in the Mediterranean area to analyse different types of possible flow dynamics in soils during flash flood events. The distributed, process-oriented model, MARINE, was used to test several representations of subsurface flows, including flows at depth in fractured bedrock, and flows through preferential pathways in macropores. Results revealed different hydrological behaviours along the catchment set, giving advances in characterising the flash flood processing over the Mediterranean area. Those results are supported by their consistency with the rare available in-situ measurements and the priori knowledge of several catchments. The characterisation is nevertheless uncompleted owing to arising equifinality issues. The descriptive potential of the distributed model was then used to spot counterbalancing effects between internal flow processes and to finally propose new insights into strategical monitoring and calibration constraints setting up."*

References would be welcome in section 1.2. References to relevant publications are added page 2, lines 16 - 24.
* * *
**Comments about the section 4:**

The description of the calibration procedure (section 4.1) and of the metrics for evaluation (section 4.2) are not sufficiently clear in my opinion, and should be improved:

- Please indicate how the "confidence intervals" are obtained for observations ($y_i \pm 2\sigma_i$?) and also for modelling results. This should clarify why the uncertainties ranges mentioned in the text (respectively $20\%$ for observations and $10\%$ for modelling errors) are consistent with eq.(6) and eq.(7). The definition of the confidence interval of the observed flows is now explicitly written page 15, lines 1 - 2 : *"The envelop $\left((\hat{y}_i \pm 2\sigma_{\hat{y}_i}),\ i = 1...n\right)$ consequently defines the 95 %*

*confidence interval of the observed flows."*

- please clarify the reason why the metric used for evaluation (Qmed_INT) is different from the one used for calibration (DEC) ? The Qmed_INT is here used as the meaning of this criteria is easier to interpret and understand, compared to the DEC value that is a standard error. This is clarified, page 16, lines 19-21: *"The DEC score has actually provided a standard assessment of the modelling errors enabling a reasonable weighting of the simulations. But in order to analyse the results, the Qmed_INT [%] score is preferred for the easy understanding it provides through it meaningful definition."*

- the definition of the "acceptability zone" should be provided ($y_i \pm 2\sigma_{y_i} \pm 2\sigma_{mod,i}$ ?). A definition of the acceptability zone is added page 15, lines 7-9 : *"Finally, the overall overarching envelop $\left( (\hat{y}_i \pm 2\sigma_{\hat{y}_i} \pm 2\sigma_{mod,i}), \ i = 1...n \right)$ defines hereafter the acceptability zone, that is to say the interval into which any simulated flow would be considered as acceptable, according to the modelling and measurement uncertainty definitions."*

- the "a priori" and "a posteriori" modelling errors are not defined. This clearly limits the interpretation of figure 11 (see hereafter). Page 17, lines 5-20 : In order to clarify the variables used in the figure 11 and the related comments, 3 paragraphs were added in the section 4.2 (Metrics and key points in model evaluation and comparison):

*The evaluation was then completed through the description of the modelling errors (section 5.2). The objective was to identify those that were inherent in the choice of model structure, regardless of the calibration methodology adopted. In that respect, attention was paid on the a priori and a posteriori confidence interval of the model simulations respectively defined by $\left( [y_i^{prior-5th}, y_i^{prior-95th}], \ i = 1...n \right)$ and $\left( [y_i^{DEC-5th}, y_i^{DEC-95th}], \ i = 1...n \right)$ where $y_i^{prior-5th}$ and $y_i^{prior-95th}$ are the $5^{th}$ and*

*the $95^{th}$ percentile of the 5000 model simulation values at time $i$, and where $y_i^{DEC-5th}$ and $y_i^{DEC-95th}$ are the $5^{th}$ and the $95^{th}$ percentile of the same but weighted series according to the DEC calibration criterion.*

*Those confidence intervals were standardized according to the DEC modelling error definition (equation 10), respectively defining the a priori and a posteriori confidence intervals of the modelling errors:*

$$\epsilon_i^{\alpha-xth} = \{\, 0 \; if \quad |\, y_i^{\alpha-xth}\,| \leq 2 \cdot \sigma_{\hat{y}_i} \frac{y_i^{\alpha-xth} \pm 2 \cdot \sigma_{\hat{y}_i}}{2 \cdot \sigma_{mod_i}} \, otherwise \quad (-\; if\; y_i^{\alpha-xth} > 0 \,; + \; if \; y_i^{\alpha-xth} \leq 0\,)$$

$$(1)$$

*with $\epsilon_i^{\alpha-xth}$ is the $x^{th}$ percentile of the $\alpha$ modelling errors distribution at time $i$.*

*The latter definition allows for an informative translation of the prior and posterior confidence intervals (Douinot et al. 2017): a value of $\epsilon_i^{\alpha-xth}$ equal to $0$ indicates that the $y_i^{\alpha-xth}$ bound lies within the discharge confidence interval; if $0 < \epsilon_i^{\alpha-xth} \leq 1$, the $y_i^{\alpha-xth}$ bound lies within the acceptability zone; and if $\epsilon_i^{\alpha-xth}$ is larger than $1$ then errors of modelling is detected or remained. In addition, the benchmark of both a priori and a posteriori confidence intervals allows for highlighting which were the remaining modelling errors that were induced by the model's assumptions. For those reasons, $\epsilon_i^{\alpha-xth}$ were used as the baseline of the modelling errors analysis.*

————————————————————————————————————————

**Comments about the section 5 (results)**

According to the comments, the presentation of the results has been reorganized as follow:

- section 5.1: Performance of the models. In these section are exclusively presented the assessment of the models through the metric scores (that are defined in the section 4.2).

- – section 5.1.1: Overall performances of the models. It merges the paragraph that has been written into the previous section 5.1.1 and 5.1.2
- – section 5.1.2: Detailed performances of the models: assessment of the models when simulating the different stages of an hydrograph. It contains the previous section 5.1.3 and 5.1.4

- • section 5.1.3: Summary of the assessment: This part has been added, in order to present a global overview of the results after detailed comments in the aforementioned sections.

- • section 5.2: Modelling errors inherent in the models'structures: It contain the previous 5.1.5 section.

- • section 5.3: Analysis of relevance of the internal hydrological processes simulated: As suggested, the previous sections 5.2 and 5.3 have been merged into one element.

- • section 6: Discussion: We propose a novel section in oder to separate the strict description of the results (section 5), and the interpretation done from it (section 6). It finally contains the previous sections 5.1.6 and 5.4.

The presentation of modelling results (section 5.1) could also be enhanced : I think the separated presentation of each metric (overall hydrograph, rising limb, high-discharges, recession) does not help to give a synthetic overview of results. It seems that three main situations can be distinguished here: clear hierarchy (Gardons and Salz), contrasted hierarchy (Ardèche), or no clear hierarchy (equi- finality, Hérault). These three situations could be illustrated based on a common analysis of all the metrics. The comments about the performances of the models were mainly reworded. In the novel version, we separated the presentation of global metric (overall hydrograph, section 5.1.1) from those that are focused on a specific stage of the hydrograph (section 5.1.2). The

objective of this organization is to highlight the differences between what we learned with a global point of view, and what we learn if we focus on the representation of one part of the hydrograph. In addition, the figures 8 and 10 were modified in order to support the new organization of the section. The figure 8 presents only the global performances while the figure 10 presents the detailed performances. Finally, for a sake of clarity, a summary is done in a last subsection (section 5.1.3). Those modifications can be find from page 18 line 1, to page 23 line 22.

Explanations in section 5.1.4 (now 5.1.2) are poorly supported by figure 10 in my opinion. Please try to clarify this section and figure. We modified the way to assess the good simulation of the flood recession, using another metric score. This is defined page 16-17 from lines 24 to line 4 :

*"Conversely, Qmed_INT was not relevant for the evaluation of the capacity to reproduce recessions, because the calculation of this score during the recession interval strongly depends on performance at high discharges. Instead, we used the $A_{slope}$ score defined in the equation 9. It calculates the average standard error in simulating the decreasing rate of the discharge during the flood recession interval. Through the consideration of the $A_{slope}$ score here, it is assumed that the recession rate is a relevant feature of the catchment's hydrologic properties Troch et al., 2013; Kirchner, 2009. We therefore choose to make a visual comparison of the simulated and observed recession curves, $Q(t) = f\left(log(-\frac{dQ(t)}{dt})\right)$, which are characteristic of a catchment's hydraulic discharge properties.*

$$A_{slope} = \frac{\sum_{i=k}^{l} |\frac{dy_i}{dt} - \frac{d\hat{y}_i}{dt}|}{\sum_{i=k}^{l} \frac{d\hat{y}_i}{dt}} \tag{2}$$

*where $\frac{d\hat{y}_i}{dt}$ and $\frac{dy_i}{dt}$ are respectively the observed and the simulated recession rate at a time step $i$ which belongs to the flood recession interval $(i = k...l)$.*

The assessment focused on the simulation of the recession is then presented in a

similar way than those on the simulation of the rising flood waters and the high flows. Consequently, the three assessments are presented in a same figure (figure 10).

The analysis proposed in section 5.1.5 (now 5.2) is also difficult to follow based on figure 11, which does not well illustrate in my opinion the differences in models behaviors. Figure 11 indeed is difficult to understand: $\sharp$ DDEC is not defined, the definition of prior and posterior errors is again missing. Moreover, is not clear why the width of the acceptability zone does not vary with $y_i$ (not consistent with equation (7)). Please try to clarify this section and figure.
In order to clarify this figure, definition of the specific variables are written in the section 4.2, page 17, lines 7-22. In addition several comments were added to better link the underlying description of the figure with the possible interpretation of the models's performances (page 23-24).

Section 5.1.6: (now 6.1) I think this interpretation on catchment behaviors arrives too early here. I think it would be better to put this in the discussion section, and to present these analyses as plausible assumptions, according to the modelling results.
As suggested, the comment of the previous section 5.1.6 are now the basis of subsection 6,1 of the discussion section.

Section 5.2 (now 5.3) may be renamed in a more explicit way, such as: "Analysis of relevance of the internal hydrological processes simulated". It could include both considerations on proportion of surface runoff (current section 5.2), and detailed analysis of velocities and water contents in the case of Hérault (current section 5.3)
As said below, we incorporated the suggestion into the new organization of the result section.

————————————————————————————————————————————

**Comments about the conclusion**

*"For each catchment, the best performing models were those where results reflected the available knowledge and observations on the overall hydrological functioning of the catchments ...".* Actually, it seems that very limited information is available on the real hydrological behavior, excepted maybe for the Gardon where detailed measurements were performed. Therefore, I would rather conclude that the modelling results help to draw consistent assumptions on hydrological behaviors, that can in some (rare) cases be confirmed by the existing knowledge and local observations.

As suggested this conclusion was reworded, taking into consideration that we actually have very limited information on those catchment (hence the interest of the study) page 35, lines 14-16 :

 *"The modelling results help to draw consistent assumptions on hydrological behaviours, which corroborate when available, the knowledge and observations on the overall hydrological functioning of the catchments, or the experimental estimations of flow processes."*

*"distinction in hydrological behaviours revealed between the catchment of the Gardons and the Ardèche may explain that taking into account the spatial nature of precipitation in a flash flood forecasting method results in an improvement only on the Gardon and not on the Ardèche ..."* I think this conclusion is not really supported by the content of the paper. Moreover, another explanation could just be a difference in the rainfall spatial variability, which seems to be more pronounced in the Gardons catchment for climatic reasons.

This last statement is introduced as an open conclusion about the potential value of the study results facing to the flash flood forecasting issue. Revealing the contrasted hydrological behaviours of the Gardons and the Ardèche catchments - the first one clearly more reactive that the second one – it might shift towards different considerations when setting up a flash flood forecasting method over those contrasted catchments. We referred to the Douinot et al, 2016 study as it actually corroborates the fact that different considerations should be done, to develop a flood forecasting method. It shows

contrasted sensitivies of the catchments to the rainfall spatial variability, which could either be a consequence of the contrasted hydrological behaviours of the catchments revealed here, or – we agree - be due to contrasted climatic forcing. We suggested to reword the statement as following (page 36, lines 1-12):

*"Lastly, identifying the most pertinent hydrological models for each catchment enables the key elements in the generation of flash floods to be highlighted, which, in turn, could serve to further develop methods for forecasting flash floods. For example, distinctions in hydrological behaviour revealed between the catchments of the Gardon and the Ardèche - the first one appearing more reactive with important runoff and subsurface flows through preferential flowpaths - might shift towards different considerations when setting up a flash flood forecasting method. It corroborates the results of Douinot et al 2016, which highlighted contrasted impacts of taking into account the spatial variability of precipitation in a flash flood forecasting method. These contrasted impacts can indeed be explained by the more pronounced spatial variability of the rainfall over the Gardon catchment, but also by the local more pronounced dynamic of the soil water content in the Gardon catchment revealed in the present study."*

———————————————————————————————————

**Technical corrections**

- Section 2.2: The reference Ministère de l'Ecologie (2015) just corresponds to an URL, which could be added directly in the text. The modification has been done (page 9, table 1 and page 10, line 2).

- "These measurements were calibrated by forecaters at the French Flood Forecasting service by monitoring a network of rain gauges ...". Sentence not clear, please reformulate. The sentence has been reworded, page 10, lines 13-15, as follow :   *"The French Flood Forecasting Service used the CALAMAR software to produce the rainfall depth inputs of the model by combining these radar mea-*

*surements with raingauge data."*

- Figure 5: are $\theta_s$ and $\theta_i$ really the current and initial water contents respectively. Shouldn't rather $\theta_s$ be the saturation water content? This is true. The error has been corrected (figure 5, page 12).

- Section 3.2, description of the modelling principles: the equations (1) to (4) and description of variables should be placed in the text with reference to figures 6 and 7. The equations and the description of the variables were inserted into the text: page 13, lines 12-15 and page 14, lines 1-8; 16.

- Section 4.2: $\sigma$ y and $\sigma$ mod rather than $\Sigma$ y and $\Sigma$ mod. The modification has been done on page 16, line 26.

- Section 5.1.5: "the variation interval of the modelling errors": I don't really understand, please define this. The expression has been reworded, page 23, line 10 as follow: *"the confidence interval of the modelling errors"*.

- I finally suggest to check the overall quality of English. The text has been proofread by a professional translator.

Please also note the supplement to this comment:
https://www.hydrol-earth-syst-sci-discuss.net/hess-2017-710/hess-2017-710-SC1-supplement.pdf

**Supplement:**

[revised manuscript text omitted]

---

## Referee Comment (RC2) · Anonymous Referee #2 · 2 Mar 2018

The article presents the test of three versions of an event-based model (MARINE) on Mediterranean catchments in France. The authors investigate the impact of the sub-surface flow and deep infiltration on model response using three modelling alternatives. They try to relate their results to the a priori knowledge on hydrological processes on the studied catchments.

First, I found that the originality of the proposed methodology is not clearly explained compared to existing works. Second, my main concern is that the results and discussion section (section 5) is excessively long and verbose. The authors discuss all the results with great detail, but the reader gets lost in all the information provided (at

[Figure]

Interactive
comment

least I got lost). At the end, it is a bit difficult to extract the main findings. I suggest reducing the size of this section to highlight the most important results. Besides, I found that the discussion on the link between results and the a priori knowledge on processes remains very qualitative. Though the explanations are sensible, there is no clear demonstration that the results are actually the consequence of the perceptual knowledge on processes invoked by the authors. There are so many possible causes to explain modelling results. I found that the reasons found by the authors only remain hypotheses and should be more clearly presented this way.

I advise major revision.

Detailed comments

1. General: Though the English writing is generally good, some sentences remain unclear. I suggest that the article be checked and corrected by native English.

2. Abstract: The main results are summarized in three lines. I find it difficult to fully understand what was done in the article by reading the abstract only.

3. Sections 1.1 and 1.2: This introduction of the context is interesting but quite classical and does not really bring essential material to understand the work done. I suggest reducing these parts to a few lines only.

4. Section 1.3: This section appears to be mostly centered on the French context. A more general perspective could be given to this literature review.

5. P4, L15-20: Clark et al. (2015a; 2015b) also proposed the SUMMA framework, applicable to distributed models. The authors should more clearly explain what is new and original in the approach they propose compared to these past works.

6. Section 2.1: This section could be presented in a more synthetic way, which would help the reader to more easily compare the study catchments. I suggest not repeating in the text information already contained in Table 1.

7. Fig. 1: For those not knowing France, maybe a small location map within France could be added.

8. Figs. 2-3-4: I suggest grouping these three figures.

9. Table 1: Be clear that QD2 and QH10 are "maximum" discharge. The HYDRO code could be introduced in the table. The meaning of Ls, L and Lsi should be made clear in the caption. Say ii the caption that bold values are dominant geology. In column ID, use the same detailed ID as those used in Table 2 for consistency. Not sure "Vogue" is the right spelling.

10. P9, L2-4: The information on flow data availability could be added in Table 1.

11. P9, L5-6: Not sure this QD2 threshold is actually the alert threshold everywhere in France. Though there may be link, I am pretty sure the alert threshold is not determined using a statistical approach, but rather by a local analysis.

12. P9, L5-11: The event selection process ignores all the rainfall events that did not generate high flows, but which would still be interesting to investigate, especially to check that the model is not over-reactive on such events. Was this analyzed in separate work? A few words could be added on this issue.

13. P9, L14: Which FFS is it?

14. P9, L14: What RHEA and CALAMR mean? Any reference?

15. P9, L26: What SIM means?

16. Section 2.2: Maybe I missed something but I did not find information on how the events were split into calibration and validation. Given there are only a few events per catchment, I guess results may be quite sensitive to this selection. This is not commented. Typically, if the authors had reversed the roles of the two events sub-sets (calibration / validation), would results be the same? If yes, this would strengthen the proposed analysis. If not, this may add further uncertainty in the analysis.

17. Table 2: For Qpeak, is it the mean of peak flows?

18. Section 3.1, title: From the description, it appears that MARINE is a model, not a framework.

19. P11, L3-10: Indicate units in brackets for parameters.

20. P11, L8: Write "Module 2 (i.e. subsurface downhill flow)"

21. P11, L12 (and elsewhere): Check the place of brackets around references.

22. P14, last line: "hourly maximum discharge"

23. P15, L17: This point was not fully clear for me. Please explain a bit more.

24. Section 5: As mentioned above, I think the authors should make an effort to much reduce this section. In several sub-sections, the comments detail so many cases that it is very difficult to get a broad picture.

25. Fig. 8: The distribution of mean results over all the catchments together would be useful to add. Is there any version that appears better on average? Please indicate in figure caption that the x axis refers to the catchments ID in Table 1.

26. Fig. 9: Colors are not very useful (especially if the article is printed black and white). Maybe use different symbols instead.

27. Section 5.2: Difficult for me to extract the main points from this long discussion...

28. Section 5.3: I was not fully convinced by the usefulness of this section.

29. Fig. 13: Are the simulation shown obtained in calibration or in validation? It would be useful to have the dates instead of the time steps on the x axis.

30. P30, L20 (and elsewhere): I think the term "demonstrate" is not appropriate. The work done here is not a demonstration. The links established between model results and actual processes remain hypotheses in the work, which may simply be more likely than others.

31. List of references: There are several incomplete references. The authors often give two URL; only keep the one for doi. Several titles are in French; please at least add the English translation in brackets, so that the non-French reader can more easily understand the topic of the cited references. I personally find it is not good practice to cite discussion papers if they were not ultimately accepted. The reference "Ministere de l'Ecologie" is strange looking.

Cited references

Clark, M. P., Nijssen, B., Lundquist, J. D., Kavetski, D., Rupp, D. E., Woods, R. A., Freer, J. E., Gutmann, E. D., Wood, A. W., Brekke, L. D., Arnold, J. R., Gochis, D. J., and Rasmussen, R. M.: A unified approach for process-based hydrologic modeling: 1. Modeling concept, Water Resour. Res., 51, 2498-2514, 10.1002/2015WR017198, 2015a.

Clark, M. P., Nijssen, B., Lundquist, J. D., Kavetski, D., Rupp, D. E., Woods, R. A., Freer, J. E., Gutmann, E. D., Wood, A. W., Gochis, D. J., Rasmussen, R. M., Tarboton, D. G., Mahat, V., Flerchinger, G. N., and Marks, D. G.: A unified approach for process-based hydrologic modeling: 2. Model implementation and case studies, Water Resour. Res., 51, 2515-2542, 10.1002/2015WR017200, 2015b.

---

## Author Comment (AC3) · 6 Apr 2018

**Using a multi-hypothesis framework to improve the understanding of flow dynamics during flash floods**

Audrey Douinot[1], Hélène Roux[1], Pierre-André Garambois[2], and Denis Dartus[1]

[1]Institut de Mécanique des Fluides de Toulouse, IMFT, Université de Toulouse, CNRS - Toulouse FRANCE
[2]Laboratoire des Sciences de l'ingénieur, de l'informatique et de l'imagerie (ICUBE) - INSA Strasbourg, Strasbourg, France
*Correspondence to:* Audrey Douinot (audreydouinot@gmail.com)

**Abstract.** A method of multiple working hypotheses was applied to a range of catchments in the Mediterranean area to analyse different types of possible flow dynamics in soils during flash flood events. The distributed, process-oriented model, MARINE, was used to test several representations of subsurface flows, including flows at depth in fractured bedrock, and flows through preferential pathways in macropores. Results showed contrasted perfomances of the submitted models, revealing different hydrological behaviours along the catchment set, and consequently, giving advances in characterising the flash flood processing over the Mediterranean area. Those results are supported by their consistency with the rare available in-situ measurements and the prior knowledge of several catchments. The characterisation is of course carried out within existing equifinality issues. The descriptive potential of the distributed model was then used 
[revised manuscript text omitted]
. But in order to analyse the results, the Qmed_INT [%] score is preferred for the easy understanding it provides through it meaningful definition. For the time interval considered, Qmed_INT defines the percentage of points within the modelling acceptability zone for the median forecast of the calibrated model, with the acceptability zone determined by $\sigma_{mod}$ et $\sigma_{\hat{y}}$.

Conversely, Qmed_INT was not relevant for the evaluation of the capacity to reproduce recessions, because the calculation
15  of this score - based on simulated discharge values - during the recession interval strongly depends on performance at high

discharges. Instead, we used the $A_{slope}$ score defined in the equation 9. It calculates the average standard error in simulating the decreasing rate of the discharge during the flood recession interval. Through the consideration of the $A_{slope}$ score here, it is assumed that the recession rate is a relevant feature of the catchment's hydrologic properties(Troch et al., 2013; Kirchner, 2009).

$$\quad A_{slope} = \frac{\sum_{i=k}^{l} |\frac{dy_i}{dt} - \frac{d\hat{y}_i}{dt}|}{\sum_{i=k}^{l} \frac{d\hat{y}_i}{dt}} \tag{9}$$

where $\frac{d\hat{y}_i}{dt}$ and $\frac{dy_i}{dt}$ are respectively the observed and the simulated recession rates at a time step $i$ which belongs to the flood recession interval $(i = k...l)$.

The evaluation was then completed through the description of the modelling errors (section 5.2). The objective was to identify those that were inherent in the choice of model structure, regardless of the calibration methodology adopted. In that

[revised manuscript text omitted]

---

## Author Response (AR1)

We wish to thank the referee for their careful evaluation of the manuscript which would be hopefully more intelligible thanks to their constructive comments. Below are our detailed responses (in blue) to the comments (in black).

Sincerely yours,
* * *
*This paper proposes a methodology for the analysis of catchment hydrological behaviors during flash floods, based on the introduction and comparison of several hypotheses in a distributed hydrological model. This topic is of broad interest for the hydrological scientific community, and is fully relevant in my opinion for a publication in HESS. However, in its current form the paper suffers from a lack of detail and explanations on several aspects (calibration procedure, explanations related to some figures, ..), causing difficulties for a detailed understanding of the research content. The presentation of sections 4 and 5 should particularly be improved in my opinion (and maybe organized in a slightly different way) to facilitate the overall understanding of the results and related analyses. The paper well illustrates the difficulties in the interpretation of modelling results, due to equifinality issues and lack of internal observations to confirm the nature of the main hydrological processes. Therefore, even if some solutions to cope with these difficulties are proposed here, I think the conclusions relative to the catchments behaviors (section 5.6) should finally be relativized and presented in the discussion section as the most reasonable assumptions, provided the modelling results obtained here.*

We are grateful for the constructive comments. Most of them shift toward quite deep modifications of the sections 4 and 5. Following the reviewer's suggestions, the organization of section 5 has been changed, and many efforts were made to describe the methodology (section 4). The answers to the comments are listed below. The modifications made in the paper are quoted in italics. Line references correspond to the unmarked manuscript (attached file). Please note also that the new version of the manuscript proposed here integrates as well the modification related to the comments of the other referee.
* * *
**Specific comments:**
The abstract is very short and could be slightly more detailed. The abstract has been reworded, giving now more details about the results (page

1, lines 5 - 10 :). Below, the reworded abstract :

*"A method of multiple working hypotheses was applied to a range of catchments in the Mediterranean area to analyse different types of possible flow dynamics in soils during flash flood events. The distributed, process-oriented model, MARINE, was used to test several representations of subsurface flows, including flows at depth in fractured bedrock, and flows through preferential pathways in macropores. Results showed contrasted perfomances of the submitted models, revealing different hydrological behaviours along the catchment set. The benchmark study offered a characterization of the catchments reactivity through the description of the hydrographs formation. The quantification of the different flow processes (surface, intra soil flows) were consistent with the scarse in-situ observations but remains uncertain, as a result of equifinality issue. The spatial description of the simulated flows over the catchments, made available by the model, enabled to spot counterbalancing effects between internal flow processes, including the compensation for the water transit time in the hillslopes and in the drainage network. New insights are finally proposed into strategical monitoring and calibration constraints setting up."*

References would be welcome in section 1.2. References to relevant publications are added page 1-2, lines 18 - 5. (note that according to the comments of the second reviewer, this section has been merge with the section 1.1 to reduce the introduction.
* * *
**Comments about the section 4:**

The description of the calibration procedure (section 4.1) and of the metrics for evaluation (section 4.2) are not sufficiently clear in my opinion, and should be improved:

- Please indicate how the "confidence intervals" are obtained for observations ($y_i \pm 2\sigma_i$?) and also for modelling results. This should clarify why the uncertainties ranges mentioned in the text (respectively 20% for observations and 10% for modelling errors) are consistent with eq.(6) and eq.(7). The definition of the confidence interval of the observed flows is now explicitly written page 12, lines 8 - 10 : *"The envelop $\left( (\hat{y}_i \pm 2\sigma_{\hat{y}_i}), \ i = 1...n \right)$ consequently defines the 95 % confidence interval of the observed flows."*

- please clarify the reason why the metric used for evaluation (Qmed_INT) is different from the one used for calibration (DEC) ? The Qmed_INT is here used as the meaning of this criteria is easier to interpret and

understand, compared to the DEC value that is a standard error. This is clarified, page 13, lines 20-25: *"While the DEC score has provided a standard assessment of the modelling errors enabling a reasonable weighting of the simulations, for a sake of easy understanding ,the percentage of acceptable points of the simulated median time series - Qmed_INT [%] (Douinot et al. 2017) - was chosen to evaluate the ability of the models to reproduce overall flows, rising flood waters and high discharges. A point is defined as acceptable when the median simulated value stands within the modelling acceptability zone, the latter one being determined by $\sigma_{mod}$ et $\sigma_{\hat{y}}$"*

- the definition of the "acceptability zone" should be provided ($y_i \pm 2\sigma_{y_i} \pm 2\sigma_{mod,i}$ ?). A definition of the acceptability zone is added page 14, lines 5-21 : *"Finally, the overall overarching envelop $\left( (\hat{y}_i \pm 2\sigma_{\hat{y}_i} \pm 2\sigma_{mod,i}),\ i = 1...n \right)$ defines hereafter the acceptability zone, that is to say the interval into which any simulated flow would be considered as acceptable, according to the modelling and measurement uncertainty definitions. "*

- the "a priori" and "a posteriori" modelling errors are not defined. This clearly limits the interpretation of figure 11 (see hereafter). Page 17, lines 5-20 : In order to clarify the variables used in the figure 11 and the related comments, 3 paragraphs were added in the section 4.2 (Metrics and key points in model evaluation and comparison):

*The evaluation was then completed through the description of the modelling errors (section 5.2). The objective was to identify those that were inherent in the choice of model structure, regardless of the calibration methodology adopted. In that respect, attention was paid on the a priori and a posteriori confidence interval of the model simulations respectively defined by $\left( [y_i^{prior-5th}, y_i^{prior-95th}],\ i = 1...n \right)$ and $\left( [y_i^{DEC-5th}, y_i^{DEC-95th}],\ i = 1...n \right)$ where $y_i^{prior-5th}$ and $y_i^{prior-95th}$ are the $5^{th}$ and the $95^{th}$ percentile of the 5000 model simulation values at time i, and where $y_i^{DEC-5th}$ and $y_i^{DEC-95th}$ are the $5^{th}$ and the $95^{th}$ percentile of the same but weighted series according to the DEC calibration criterion.*

*Those confidence intervals were standardized according to the DEC modelling error definition (equation 10), respectively defining the a priori and a posteriori confidence intervals of the modelling errors:*

$$\epsilon_i^{\alpha-xth} = \begin{cases} 0 & if \quad | y_i^{\alpha-xth} | \leq 2 \cdot \sigma_{\hat{y}_i} \\ \frac{y_i^{\alpha-xth} \pm 2 \cdot \sigma_{\hat{y}_i}}{2 \cdot \sigma_{mod_i}} & otherwise \quad (-\ if\ y_i^{\alpha-xth} > 0\,;\ +\ if\ y_i^{\alpha-xth} \leq 0\,) \end{cases}$$

*with $\epsilon_i^{\alpha-xth}$ is the $x^{th}$ percentile of the $\alpha$ modelling errors distribution at time i.*

*The latter definition allows for an informative translation of the prior and posterior confidence intervals (Douinot et al. 2017): a value of $\epsilon_i^{\alpha-xth}$ equal to 0 indicates that the $y_i^{\alpha-xth}$ bound lies within the discharge confidence interval; if $0 < \epsilon_i^{\alpha-xth} \leq 1$, the $y_i^{\alpha-xth}$ bound lies within the acceptability zone; and if $\epsilon_i^{\alpha-xth}$ is larger than 1 then errors of modelling is detected or remained. In addition, the benchmark of both a priori and a posteriori confidence intervals allows for highlighting which were the remaining modelling errors that were induced by the model's assumptions. For those reasons, $\epsilon_i^{\alpha-xth}$ were used as the baseline of the modelling errors analysis.*
* * *
**Comments about the section 5 (results)**

According to the comments, the presentation of the results has been reorganized as follow:

- section 5.1: Performance of the models. In these section are exclusively presented the assessment of the models through the metric scores (that are defined in the section 4.2).

  - section 5.1.1: Overall performances of the models. It merges the paragraph that has been written into the previous section 5.1.1 and 5.1.2

  - section 5.1.2: Detailed performances of the models: assessment of the models when simulating the different stages of an hydrograph. It contains the previous section 5.1.3 and 5.1.4

- section 5.1.3: Summary of the assessment: This part has been added, in order to present a global overview of the results after detailed comments in the aforementioned sections.

- section 5.2: Modelling errors inherent in the models'structures: It contain the previous 5.1.5 section.

- section 5.3: Analysis of relevance of the internal hydrological processes simulated: As suggested, the previous sections 5.2 and 5.3 have been merged into one element.

- section 6: Discussion: We propose a novel section in oder to separate the strict description of the results (section 5), and the interpretation done from it (section 6). It finally contains the previous sections 5.1.6 and 5.4.

The presentation of modelling results (section 5.1) could also be enhanced : I think the separated presentation of each metric (overall hydrograph, rising limb, high-discharges, recession) does not help to give a synthetic overview of results. It seems that three main situations can be distinguished here: clear hierarchy (Gardons and Salz), contrasted hierarchy (Ardche), or no clear hierarchy (equi- finality, Hrault). These three situations could be illustrated based on a common analysis of all the metrics. The comments about the performances of the models were mainly reworded. In the novel version, we separated the presentation of global metric (overall hydrograph, section 5.1.1) from those that are focused on a specific stage of the hydrograph (section 5.1.2). The objective of this organization is to highlight the differences between what we learned with a global point of view, and what we learn if we focus on the representation of one part of the hydrograph. In addition, the figures 8 and 10 were modified in order to support the new organization of the section. The figure 8 (now figure 6, page 15) presents only the global performances while the figure 10 (now figure 8, page 17) presents the detailed performances. Finally, for a sake of clarity, a summary is done in a last subsection (section 5.1.3). Those modifications can be find from page 14 line 22, to page 18 line 10.

Explanations in section 5.1.4 (now 5.1.2) are poorly supported by figure 10 in my opinion. Please try to clarify this section and figure. We modified the way to assess the good simulation of the flood recession, using another metric score. This is defined page 14 from lines 25 to line 5 :

*"Conversely, Qmed_INT was not relevant for the evaluation of the capacity to reproduce recessions, because the calculation of this score during the recession interval strongly depends on performance at high discharges. Instead, we used the $A_{slope}$ score defined in the equation 9. It calculates the average standard error in simulating the decreasing rate of the discharge during the flood recession interval. Through the consideration of the $A_{slope}$ score here, it is assumed that the recession rate is a relevant feature of the catchment's hydrologic properties Troch et al., 2013; Kirchner, 2009. We therefore choose to make a visual comparison of the simulated and observed recession curves, $Q(t) = f\left(log\left(-\frac{dQ(t)}{dt}\right)\right)$, which are characteristic of a catchment's hydraulic discharge properties.*

$$A_{slope} = \frac{\sum_{i=k}^{l} |\frac{dy_i}{dt} - \frac{d\hat{y}_i}{dt}|}{\sum_{i=k}^{l} \frac{d\hat{y}_i}{dt}} \tag{1}$$

*where $\frac{d\hat{y}_i}{dt}$ and $\frac{dy_i}{dt}$ are respectively the observed and the simulated recession rate at a time step i which belongs to the flood recession interval $(i = k...l)$.*

The assessment focused on the simulation of the recession is then presented in a similar way than those on the simulation of the rising flood waters and

the high flows. Consequently, the three assessments are presented in a same figure (figure 8).

The analysis proposed in section 5.1.5 (now 5.2) is also difficult to follow based on figure 11, which does not well illustrate in my opinion the differences in models behaviors. Figure 11 indeed is difficult to understand: ♯ DDEC is not defined, the definition of prior and posterior errors is again missing. Moreover, is not clear why the width of the acceptability zone does not vary with $y_i$ (not consistent with equation (7)). Please try to clarify this section and figure.
In order to clarify this figure (figure 10 now), definition of the specific variables are written in the section 4.2, page 14, lines 6-21. In addition several comments were added to better link the underlying description of the figure with the possible interpretation of the modelss performances (page 18-19).

Section 5.1.6: (now 6.1) I think this interpretation on catchment behaviors arrives too early here. I think it would be better to put this in the discussion section, and to present these analyses as plausible assumptions, according to the modelling results.
As suggested, the comment of the previous section 5.1.6 are now the basis of subsection 6,1 of the discussion section.

Section 5.2 (now 5.3) may be renamed in a more explicit way, such as: "Analysis of relevance of the internal hydrological processes simulated". It could include both considerations on proportion of surface runoff (current section 5.2), and detailed analysis of velocities and water contents in the case of Hérault (current section 5.3)
As said below, we incorporated the suggestion into the new organization of the result section.
* * *
**Comments about the conclusion**

*"For each catchment, the best performing models were those where results reflected the available knowledge and observations on the overall hydrological functioning of the catchments ..."*. Actually, it seems that very limited information is available on the real hydrological behavior, excepted maybe for the Gardon where detailed measurements were performed. Therefore, I would rather conclude that the modelling results help to draw consistent assumptions on hydrological behaviors, that can in some (rare) cases be confirmed by the existing knowledge and local observations.
As suggested this conclusion was reworded, taking into consideration that we actually have very limited information on those catchment (hence the

interest of the study) page 29, lines 19-21 :

*"The modelling results help to draw consistent assumptions on hydrological behaviours, which corroborate when available, the knowledge and observations on the overall hydrological functioning of the catchments, or the experimental estimations of flow processes."*

*"distinction in hydrological behaviours revealed between the catchment of the Gardons and the Ardèche may explain that taking into account the spatial nature of precipitation in a flash flood forecasting method results in an improvement only on the Gardon and not on the Ardèche ..."* I think this conclusion is not really supported by the content of the paper. Moreover, another explanation could just be a difference in the rainfall spatial variability, which seems to be more pronounced in the Gardons catchment for climatic reasons.

This last statement is introduced as an open conclusion about the potential value of the study results facing to the flash flood forecasting issue. Revealing the contrasted hydrological behaviours of the Gardons and the Ardche catchments - the first one clearly more reactive that the second one it might shift towards different considerations when setting up a flash flood forecasting method over those contrasted catchments. We referred to the Douinot et al, 2016 study as it actually corroborates the fact that different considerations should be done, to develop a flood forecasting method. It shows contrasted sensitivities of the catchments to the rainfall spatial variability, which could either be a consequence of the contrasted hydrological behaviours of the catchments revealed here, or we agree - be due to contrasted climatic forcing. We suggested to reword the statement as following (page 30, lines 4-11):

*"Lastly, identifying the most pertinent hydrological models for each catchment enables the key elements in the generation of flash floods to be highlighted, which, in turn, could serve to further develop methods for forecasting flash floods. For example, distinctions in hydrological behaviour revealed between the catchments of the Gardon and the Ardèche - the first one appearing more reactive with important runoff and subsurface flows through preferential flowpaths - might shift towards different considerations when setting up a flash flood forecasting method. It corroborates the results of Douinot et al 2016, which highlighted contrasted impacts of taking into account the spatial variability of precipitation in a flash flood forecasting method. These contrasted impacts can indeed be explained by the more pronounced spatial variability of the rainfall over the Gardon catchment, but also by the local more pronounced dynamic of the soil water content in the Gardon catchment revealed in the present study."*

**Technical corrections**

- Section 2.2: The reference Ministre de lEcologie (2015) just corresponds to an URL, which could be added directly in the text. The modification has been done (page 6, table 1 and page 7, line 3).

- "These measurements were calibrated by forecaters at the French Flood Forecasting service by monitoring a network of rain gauges ...". Sentence not clear, please reformulate. The sentence has been reworded, page 17, lines 15-18, as follow : *"The French flood forecasting service (SCHAPI: Service central d'hydrométéorologie et d'appui à la prévision des inondations) used then the CALAMAR patented software (Badoche-Jacquet et al. 1992) to produce rainfall depth data by combining these radar measurements with raingauge data."*

- Figure 5 (now figure 3): are $\theta_s$ and $\theta_i$ really the current and initial water contents respectively. Shouldnt rather $\theta_s$ be the saturation water content? This is true. The error has been corrected (figure 3, page 9).

- Section 3.2, description of the modelling principles: the equations (1) to (4) and description of variables should be placed in the text with reference to figures 6 and 7. The equations and the description of the variables were inserted into the text: page 10, lines 15-20 and page 11, lines 7-15; 21.

- Section 4.2: y and mod rather than y and mod. The modification has been done on page 13, line 24.

- Section 5.1.5: the variation interval of the modelling errors: I dont really understand, please define this. The expression has been reworded, page 19, line 5 as follow: *"the width of the confidence interval of the modelling"*.

- I finally suggest to check the overall quality of English. The text has been proofread by a professional translator.

We wish to thank the referee for his careful evaluation of the manuscript. Please find below, the details responses (in blue) to the comments (in black). Please note also that the new version of the manuscript proposed here integrates as well the modification related to the comments of the other referee. The mentioned line refer to the unmarked manuscript.

Sincerely yours,

Audrey Douinot

———————————————————————————————————————————————

*The article presents the test of three versions of an event-based model (MARINE) on Mediterranean catchments in France. The authors investigate the impact of the subsurface flow and deep infiltration on model response using three modelling alternatives. They try to relate their results to the a priori knowledge on hydrological processes on the studied catchments. First, I found that the originality of the proposed methodology is not clearly explained compared to existing works. Second, my main concern is that the results and discussion section (section 5) is excessively long and verbose. The authors discuss all the results with great detail, but the reader gets lost in all the information provided (at least I got lost). At the end, it is a bit difficult to extract the main findings. I suggest reducing the size of this section to highlight the most important results. Besides, I found that the discussion on the link between results and the a priori knowledge on processes remains very qualitative. Though the explanations are sensible, there is no clear demonstration that the results are actually the consequence of the perceptual knowledge on processes invoked by the authors. There are so many possible causes to explain modelling results. I found that the reasons found by the authors only remain hypotheses and should be more clearly presented this way.*

The general comments are mostly related to the unclear presentation of the results and the insights of the work. As those comments are quite similar to the those from the other referee, deep modifications have been done on the last sections (section 5 and the conclusion):

- the section 5 has been splitted into a section "results" that exactly describes the results when applying the method presented in section 4; and a section "discussion" where those results are interpreted, as the witnesses of the hydrological behavior of the catchment set.

- the conclusion was reworded to clarify which insights are clearly demonstrated from those that are proposals, that still need to be checked with additional field observations.

———————————————————————————————————————————————

**Specific comments:**

1. General: Though the English writing is generally good, some sentences remain unclear. I suggest that the article be checked and corrected by native English.

The text has been proofread by a professional translator, native english.

2. Abstract: The main results are summarized in three lines. I find it difficult to fully understand what was done in the article by reading the abstract only.

The abstract has been reworded, giving now more details about the results (page 1, lines 5 - 10). Below, the reworded abstract :

*"A method of multiple working hypotheses was applied to a range of catchments in the Mediterranean area to analyse different types of possible flow dynamics in soils during flash flood events.*

*The distributed, process-oriented model, MARINE, was used to test several representations of sub-surface flows, including flows at depth in fractured bedrock, and flows through preferential pathways in macropores. Results showed contrasted perfomances of the submitted models, revealing different hydrological behaviours along the catchment set. The benchmark study offered a characterization of the catchments reactivity through the description of the hydrographs formation. The quantification of the different flow processes (surface, intra soil flows) were consistent with the scarse in-situ observations but remains uncertain, as a result of equifinality issue. The spatial description of the simulated flows over the catchments, made available by the model, enabled to spot counterbalancing effects between internal flow processes, including the compensation for the water transit time in the hillslopes and in the drainage network. New insights are finally proposed into strategical monitoring and calibration constraints setting up."*

3. Sections 1.1 and 1.2: This introduction of the context is interesting but quite classical and does not really bring essential material to understand the work done. I suggest reducing these parts to a few lines only.

The sections 1.1 and 1.2 - that is to say the beginning of the article - has been reduced as follow (page 1 and 2, from lines 15 to 26). Below, the reworded introduction:

*Flash floods are "sudden floods with high peak discharges, produced by severe thunderstorms that are generally of limited areal extent". (IAHS-UNESCO-WMO (1974); Garambois (2012); Braud et al. (2014)). They are often linked to localised and major forcings (greater than 100 mm, Gaume et al. (2009)) at the heads of steep-sided, meso-scale catchments (with surface areas of 10-250 $km^2$).*

*The large specific discharges, and intensities of precipitation, makes the flash floods being classified as extreme. Nevertheless, those events are not scarce nor unusual since on average, there were no fewer than five flash floods a year on the Mediterranean Arc between 1958 and 1994 (Jacq, 1994), and they tend to be amplifed against a background of climate change (Llasat et al., 2014; Colmet Daage et al., 2016). Flash floods constitute a significant hazard and, therefore, a considerable risk for populations (UNISDR 2009, Llasat et al. (2014)). They are particularly dangerous due to their characteristics: (i) the suddenness of events makes it difficult to warn populations in time, and can lead to panic, thus increasing risk, when a population is unprepared (Ruin et al., 2008); ii) the traditional connected monitoring system are not adapted to the temporal and spatial scales of the flash floods (Borga et al., 2008; Braud et al., 2014); iii) the magnitude of floods implies significant amounts of kinetic energy, which can transform transitory rivers into torrents, resulting in the transport of debris ranging from fine sediments to tree trunks, as well as the scouring of river beds and the erosion of banks (Borga et al., 2014).*

*A major area of interest for flash floods is, therefore, better risk assessment, to enable them to be forecast and the relevant populations to be pre-warned. Greater knowledge and understanding is required to better identify the determining factors that result in flash floods. In particular, in order to implement a regional forecasting system, the properties of the catchments, and the climatic forcing and linkages between them which lead to flash flood events need to be characterised.*

4. Section 1.3 (now 1.2): This section appears to be mostly centered on the French context. A more general perspective could be given to this literature review.

This is true. The bibliography here was actually quite consciously centered to the North-West Mediterranean context, as behind the "flash flood" term, there are different types of hydro-meteorological events, basically depending on the area over the world concerned. As example in Europe, events from the South East of France, Northern Italy and North Eastern Spain (Catalonia) are characterized by more intensive rainfall during larger time extents, and exhibit different climatic conditions than the flash floods occuring in the east part of Europe (along the Carpathian

mountains and the continental Alps region, Tarolli et al. (2012)). As our study is specially focused on catchments of the French Mediterranean area, the bibliography similarly took interest in flash flood events with similar meteorological context. Nevertheless, some references has been added to offer a first broader overview over the current researches on the topic, and to complete the statements found in the litterature on flash flood processing over the North-West Meditterranean area. This focus, on this particular content has also been specified (page 2, lines 21-22).

5. P4, L15-20: Clark et al. (2015a; 2015b) also proposed the SUMMA framework, applicable to distributed models. The authors should more clearly explain what is new and original in the approach they propose compared to these past works.

The proposed study relies on the cited approaches. The originality is to apply the approach on an event-based model, that - as far as we know - has not been done before. The objective here is to bring insights on hydrological understanding in a specific case (flash flood processing in the North-Westhern Mediterranean area) using an alike promising approach described in the past works. I added the reference you suggested, as clearly missing (page 3, line 24-29). In addition, the bold font, highlighting the objectives of the paper has been modified (It would have lead to confusion).

6. Section 2.1: This section could be presented in a more synthetic way, which would help the reader to more easily compare the study catchments. I suggest not repeating in the text information already contained in Table 1.

The description of the catchments has been reduced to the description of the contrasted geology - as the most significant information with the objective of the study, and to the current knowlege on the hydrological processes of the studied area obtained through field experimental studies. You will find the reworded subsection in page 4-5, lines 15-15, or here below:

*The main physiographical and hydrological properties of the catchments are presented in Table 1. Figure 2 shows the contrasted geological properties of the studied area : the catchments are marked by a clear upstream / downstream difference. The Ardèche catchment upstream of Ucel sits essentially on a granite bedrock with some sandstone on its edges, while downstream, the geology changes to a predominantly schist and limestone formations. Similarly, the upstream part of the Gardon catchment consists of schistose bedrock while, downstream, the bedrock is impermeable marl-type and granite formation. The Herault catchment is splited into mostly schist and granitic head watersheds (the Valleraugue and la Terrisse sub-catchments) and a predominantly limestone plateau (Saint Laurent le Minier sub-catchment). Finally, the Salz is characterised by sedimentary bedrock comprising sandstone and limestone (Figure 2).*

*The Ardèche and the Gardon catchments have been subject to intensive monitoring and studies (see lter reference, https://data.lter-europe.net/deims/site/rbv_fr_13), leading to prior knowledge on hydrological understanding. Both the local in-situ experiments (Ribolzi et al., 1997; Braud and Vandervaere, 2015; Braud et al., 2016a,b) and the modelling studies focused on this area (Garambois et al., 2013; Vannier et al., 2013) tend to a hydrological classification according to those contrasted geological properties and, in agreements with the usual hydrogeological signature found in the litterature (Sayama et al., 2011; Pfister et al., 2017). Marls, sandstone and limestones without karst are characterized by limited storage capacities, resulting in higher runoff coefficients, and high sensitivity to the initial soil moisture (Ribolzi et al., 1997; Braud et al., 2016a). In contrast, in granite and schist transects located on hillslope of the Ardèche catchment, infiltration tests and analysis of electrical resisitivity signals show high permeability of the geological substratum in depth (measured up to 2.5 m in depth); and high storage capacities reaching up to 600 mm in 7 out of 10 assessments with artificial forcing, the 3 remaining test suggesting local unaltered bedrock (Braud et al., 2016a,b). The natural resistivity profile suggests a regular soil bedrock interface*

*when the latter consist in schist, while the granite one presents a more chaotic structure. Finally, the continous comparative study of two experimental sites over surface areas of the order of one km² - one located on the schist upstream part of the Gardon catchment, the other one on it granite downstream part - suggests rapid subsurface flow processing on the schist area, while flow formation appears to be controlled by the extension of the saturated zone related to the river on the granitic site (Ayral et al., 2005; Maréchal et al., 2009, 2013).*

7. Fig. 1: For those not knowing France, maybe a small location map within France could be added.

The France map has been added.

8. Figs. 2-3-4: I suggest grouping these three figures.

We followed your suggestion and grouped the figures.

9. Table 1: Be clear that QD2 and QH10 are maximum discharge. The HYDRO code could be introduced in the table. The meaning of Ls, L and Lsi should be made clear in the caption. Say ii the caption that bold values are dominant geology. In column ID, use the same detailed ID as those used in Table 2 for consistency. Not sure Vogue is the right spelling.

The missing caption on the table 1 have been added, and the table has been reorganized (the columns order has been changed to consistency group the properties of the catchments. Finally, the outlet "Vogue" has been changed to the correct French spelling "Vogüé", here in the table and in all the manuscript. Hereafter, the new caption of the table 1:

*Physiographic properties and hydrological statistics of the 12 catchments ID: coding name of the catchments used at figure 1 and table 2; area [km²]; mean slope [-]; soil properties: mean soil depth [m] and main soil texture (Tx) : Ls = sandy loam texture, L = loam texture; Lsi = silty loam texture; Geology: percentage of bedrock geology [%] including sandstone (Sa), limestone (Li), granite and gneiss (GG), marls (Ma) and schists (Sc) subcategories - $^{(i)}$ bold values are the dominant geology; mean annual precipitation (P[mm]) ; Hydrometry: discharge time-series availability (Period); mean inter-annual discharge (Q[m³.km⁻².s⁻¹]); 2 year return period of maximum daily discharge (Q_{D2}[m³.km⁻².s⁻¹]); 10 year return period of maximum hourly discharge (Q_{H10}[m³.km⁻².s⁻¹]). Hydrometric statistics are calculated from HydroFrance databank, (de l'Ecologie du développement durable et de l'énergie, 2015) (http://www.hydro.eaufrance.fr/) and the pluviometric ones using rainfall data from the raingauge network of the French flood forecasting services.*

10. P9, L2-4: The information on flow data availability could be added in Table 1.

The flow data availability have been added in Table 1.

11. P9, L5-6: Not sure this QD2 threshold is actually the alert threshold everywhere in France. Though there may be link(ed), I am pretty sure the alert threshold is not determined using a statistical approach, but rather by a local analysis.

The flood warning system in France, had actually been restrustured, in the beginning of the 00's, after dramatic consequences of several flash flood events in 1999 and 2002. The main objective was to improve not only the flood forecasts, but also the communication with local authorities. To meet that objective, a subdaily flood warning map of the main rivers in France is broadcoast through a unique website (https://www.vigicrues.gouv.fr/). For a sake of clarity in the communication, a uniform color code is used: yellow for peak discharge ranging from the 2-year to the 10- year flood, orange for peak discharge ranging from the 10-year to the 50-year flood, and red for peak

discharge exceeding the 50-year flood. I didn't find any reference in english, but you can find some description in (Javelle et al., 2014): *In real-time, to describe the potential severity of the ongoing event along the river network, the estimated peak discharges are represented with a colour code based on three flood frequency categories: yellow for peak discharge ranging from the 2-year to the 10- year flood, orange for peak discharge ranging from the 10-year to the 50-year flood, and red for peak discharge exceeding the 50-year flood. These real-time products, delivered every 15 min, are used as input for a web site dedicated to French local authorities.*

12. P9, L5-11: The event selection process ignores all the rainfall events that did not generate high flows, but which would still be interesting to investigate, especially to check that the model is not over-reactive on such events. Was this analyzed in separate work? A few words could be added on this issue.

Those events have been not studied yet as the model has not been built to simulate this kind of responses. For instance there is no percolation and no groundwater recharge. When looking for the assessment of the models as a flood forecast tool, those event could have be to integrated on the study. Here, looking at the flow processes, they were not included, the precipitation being mostly infiltrated. Also, as the discharge threshold is moderate (2 years return period of the maximal daily discharge), the selection already includes intense rainfall events, with only moderate hydrological response, as suggest the different runoff coefficients of the events. Consequently we added this statistic on the table 2 (page 8).

The following sentence has also been added page 7, lines 10-14: *The aim of this selection was to be able to analyse, more broadly, overall catchment behaviour during intense  hydrological activity. Note also that, moderate or intense rainfall events without respective hydrological response might be abducted from the analysis. Nevertheless the first alert threshold used here is small enough to have a selection of flood events with contrasted runoff coefficient (see table 2.)*

13. P9, L14: Which FFS is it?

Here, it specially concerns two regional flood forecasting services (SPC): the SPC Grand Delta and the SPC Med-Ouest. However, to be synthetic, we refer to the national French flood forecasting service SCHAPI that is at the head of the regional ones (page 9, line 15).

14. P9, L14: What RHEA and CALAMR mean? Any reference?

CALAMAR is a patented software developed by a private company RHEA. The sentence has been reworded to be more understandable and to add the reference of the patent (page 9, lines 14-16). The reworded sentence is: *The French flood forecasting service (SCHAPI: Service central dhydrométéorologie et dappui à la prévision des inondations) used the CALAMAR patented software (Badoche-Jacquet et al., 1992) to produce the rainfall depth inputs of the model by combining these radar measurements with raingauge data.*

15. P9, L26: What SIM means?

SIM for the models used into the operational chain : the Safran model, a meteorological analysis system; the ISBA model simulated the Interaction between the Soil, the Biosphere, and the Atmosphere; and the Modcou model, a hydrogeological model. Those details have been succintly added page 8, lines 4-6:

*This was done using spatial model outputs from Météo-France's SIM operational chain (Habets et al., 2008), including a meterological model (SAFRAN, Vidal et al. (2010)), a soil - vegetation - atmosphere model (ISBA, Mahfouf et al. (1995)) and a hydrogeological model (MODCOU, Ledoux et al. (1989)).*

16. Section 2.2: Maybe I missed something but I did not find information on how the events were split into calibration and validation. Given there are only a few events per catchment, I guess results may be quite sensitive to this selection. This is not commented. Typically, if the authors had reversed the roles of the two events sub-sets (calibration / validation), would results be the same? If yes, this would strengthen the proposed analysis. If not, this may add further uncertainty in the analysis.

Here, each event set was splitted into calibration and validation sets, according to the work of Garambois et al. (2015). As suggested in the latter cited paper, a first individual calibration on each event was done. Events presenting atypical sensivity to the soil depth parameter, has been removed from the calibration. The extreme events were kept for validation. And finally, events were splitted in order to have a wide range of soil moisture initial condition. The following sentences have been added to detail the event set splitting (page 7-8, lines 18-2):

*Each rainfall product is firstly assessed through an individual sensitivity analysis of the standard MARINE model (DWF model, see section 3.1). When presenting an atypical sensivity to the soil depth parameter, the rainfall event is discarded of the study, as suggesting questionable measurements. Depending on the availability of the results of rainfall and hydrometric measurements, 7 to 14 intense events were selected for each catchment (Table 2). Each set is finally splitted into a calibration and validation subsets as follow: the extreme events were kept for validation. A minimum number of calibration events is chosen in order to cover the wide range of soil moisture initial condition.*

17. Table 2: For Qpeak, is it the mean of peak flows?

Exactly, the caption has been corrected.

18. Section 3.1, title: From the description, it appears that MARINE is a model, not a framework.

The title of the section 3.1 has been modified.

19. P11, L3-10: Indicate units in brackets for parameters.

The parenthesis have been replaced for brackets.

20. P11, L8: Write Module 2 (i.e. subsurface downhill flow)

The specification has been added.

21. P11, L12 (and elsewhere): Check the place of brackets around references.

Ok. Brackets around references have been check in the proofreading.

22. P14, last line: hourly maximum discharge

Ok.

23. P15, L17: This point was not fully clear for me. Please explain a bit more.

A different mean of evaluation is used for the hydrograph recession as the assessement based on the simulated discharge values - Qmed_INT - and restricted to the recession interval is actually representative whether the high discharge values occuring before the recession are well simulated or not. For this reason, to assess the hydrograph recession, a score based on the recession rate is rather used, as it enables to avoid such a dependency of the assessement with the high discharge

values, but also as it is very well known in the litterature as representative of the hydrological behavior of the catchment.

According to your comments and those from the other referee, another assessment is proposed in the reviewed manuscript. Instead of having a special figure for the assessment of the hydrograph recession, the detailed performances over the different stages are grouped in a same figure (figure 8). To meet that objective, a novel score based on the recession rate is proposed and used to evaluate the simulations of the related stage of the hydrographs. You can find these modifications on the section 4.2,page 13-14, lines 25-5:

*Conversely, Qmed_INT  was not relevant for the evaluation of the capacity to reproduce recessions, because the calculation of this score - based on simulated discharge values - during the recession interval strongly depends on performance at high discharges. Instead, we used the $A_{slope}$ score defined in the equation 9. It calculates the average standard error in simulating the decreasing rate of the discharge during the flood recession interval. Through the consideration of the $A_{slope}$ score here, it is assumed that the recession rate is a relevant feature of the catchment's hydrologic properties (Troch et al., 2013; Kirchner, 2009). ~~We therefore choose to make a visual comparison of the simulated and observed recession curves, $Q(t) = f\left(log\left(-\frac{dQ(t)}{dt}\right)\right)$, which are characteristic of a catchment's hydraulic discharge properties. Lastly, the evaluation was completed by a description of the a priori and a posteriori modelling errors in order to identify those that were inherent in the choice of model structure, regardless of the calibration strategy adopted.~~*

$$A_{slope} = \frac{\sum_{i=k}^{l} \left|\frac{dy_i}{dt} - \frac{d\hat{y}_i}{dt}\right|}{\sum_{i=k}^{l} \frac{d\hat{y}_i}{dt}} \tag{1}$$

*where $\frac{d\hat{y}_i}{dt}$ and $\frac{dy_i}{dt}$ are respectively the observed and the simulated recession rate at a time step i which belongs to the flood recession interval $(i = k...l)$.*

24. Section 5: As mentioned above, I think the authors should make an effort to much reduce this section. In several sub-sections, the comments detail so many cases that it is very difficult to get a broad picture.

According to your comments and those from the other referee, the presentation of the results has been reorganized as follow:

- section 5.1: Performance of the models. In these section are exclusively presented the assessment of the models through the metric scores (that are defined in the section 4.2).
    - section 5.1.1: Overall performances of the models. It merges the paragraph that has been written into the previous section 5.1.1 and 5.1.2
    - section 5.1.2: Detailed performances of the models: assessment of the models when simulating the different stages of an hydrograph. It contains the previous section 5.1.3 and 5.1.4
- section 5.1.3: Summary of the assessment: This part has been added, in order to present a global overview of the results after detailed comments in the aforementioned sections.
- section 5.2: Modelling errors inherent in the models'structures: It contain the previous 5.1.5 section.
- section 5.3: Analysis of relevance of the internal hydrological processes simulated: As suggested, the previous sections 5.2 and 5.3 have been merged into one element.

- section 6: Discussion: We propose a novel section in oder to separate the strict description of the results (section 5), and the interpretation done from it (section 6). It finally contains the previous sections 5.1.6 and 5.4.

25. Fig. 8: The distribution of mean results over all the catchments together would be useful to add. Is there any version that appears better on average? Please indicate in figure caption that the x axis refers to the catchments ID in Table 1.

The distribution of mean results over all the catchments together has been added and the caption was modified.

26. Fig. 9: Colors are not very useful (especially if the article is printed black and white). Maybe use different symbols instead.

Suggested modification were done.

27. Section 5.2: Difficult for me to extract the main points from this long discussion...

The section 5.2 (now 5.3.1) has totally been reworded (page 21).

28. Section 5.3: I was not fully convinced by the usefulness of this section.

The results of the section 5.3 (now 5.3.2) follows the section 5.3.1 about the assessment of the proportional volumes of the water up the hydrographs, that arise from the thre main path : on the surface, through the top or the deep layer of the soil. While those assessements are incomplete because of large uncertainties, the section 5.3.2 details their origins, revealing how the different models can involve at some points different internal dynamics and how two parameter sets can lead to simulate similar hydrographs, allowing a wide range of velocities, and then counterbalancing transfer time offsets between internal flow processes. We kept this section as, in our opinion, being relevant for the benchmark of the models, and interesting insights for further studies.

29. Fig. 13: Are the simulation shown obtained in calibration or in validation? It would be useful to have the dates instead of the time steps on the x axis.

As suggested, the x axis has been modified. Among the simulation shown, three are calibration events and one is a validation event. This has been added in the caption. In the same way, this detail has been also added one figure 10.

30. P30, L20 (and elsewhere): I think the term demonstrate is not appropriate. The work done here is not a demonstration. The links established between model results and actual processes remain hypotheses in the work, which may simply be more likely than others.

The term "demonstrate" was replaced by "indicate".

31. List of references: There are several incomplete references. The authors often give two URL; only keep the one for doi. Several titles are in French; please at least add the English translation in brackets, so that the non-French reader can more easily understand the topic of the cited references. I personally find it is not good practice to cite discussion papers if they were not ultimately accepted. The reference Ministere de lEcologie is strange looking.

The list of references has been checked and we propose a translation for the French cited papers.

**References**

[revised manuscript text omitted]
 1 : $\sharp$1 for the Ardèche; $\sharp$2 for the Gardon; $\sharp$3 for the Hérault and $\sharp$4 for the Salz); $N_{evt}$: number of observed flash flood events; P [mm] mean precipitation ; $I_{max}[mm.h^{-1}]$: maximal intensity rainfall per event; $Q_{peak}$: specific flood peak $[m^3.km^{-2}.s^{-1}]$; Hum: initial soil moil moisture according to SIM output (Habets et al., 2008); CR: runoff coeficient [%]

[revised manuscript text omitted]

---

## Author Response (AR2)

Dear editor,

We revised the manuscript, correcting the typo/english errors when found, and making modifications mainly in sections 5, 6 and 7 as suggested. They mainly consisted in removing some repetitions and clarifying the explanations. Please, find the marked (as supplement) and not marked (as manuscript) revised manuscript.

Best regards,

Audrey Douinot

[revised manuscript text omitted]